# BRN2 is a non-canonical melanoma tumor-suppressor

Michael Hamm[1,2,3,13], Pierre Sohier [1,2,3,13], Valérie Petit[1,2,3,13], Jérémy H. Raymond[1,2,3], Véronique Delmas [1,2,3], Madeleine Le Coz[1,2,3], Franck Gesbert[1,2,3], Colin Kenny[4], Zackie Aktary[1,2,3], Marie Pouteaux[1,2,3], Florian Rambow[1,2,3], Alain Sarasin[5], Nisamanee Charoenchon[1,2,3,6], Alfonso Bellacosa[7], Luis Sanchez-del-Campo [8], Laura Mosteo[8], Martin Lauss[9], Dies Meijer [10], Eirikur Steingrimsson[11], Göran B. Jönsson[9], Robert A. Cornell[4], Irwin Davidson [4,12], Colin R. Goding [8✉] & Lionel Larue [1,2,3✉]

While the major drivers of melanoma initiation, including activation of NRAS/BRAF and loss of *PTEN* or *CDKN2A*, have been identified, the role of key transcription factors that impose altered transcriptional states in response to deregulated signaling is not well understood. The POU domain transcription factor BRN2 is a key regulator of melanoma invasion, yet its role in melanoma initiation remains unknown. Here, in a *Braf*$^{V600E}$ *Pten*$^{F/+}$ context, we show that *BRN2* haplo-insufficiency promotes melanoma initiation and metastasis. However, metastatic colonization is less efficient in the absence of Brn2. Mechanistically, BRN2 directly induces *PTEN* expression and in consequence represses PI3K signaling. Moreover, MITF, a BRN2 target, represses *PTEN* transcription. Collectively, our results suggest that on a *PTEN* heterozygous background somatic deletion of one *BRN2* allele and temporal regulation of the other allele elicits melanoma initiation and progression.

[1] Institut Curie, Université PSL, CNRS UMR3347, Inserm U1021, Normal and Pathological Development of Melanocytes, Orsay, France. [2] Université Paris-Saclay, CNRS UMR3347, Inserm U1021, Signalisation radiobiologie et cancer, Orsay, France. [3] Equipes Labellisées Ligue Contre le Cancer, Paris, France. [4] Department of Anatomy and Cell biology, Carver College of Medicine, University of Iowa, Iowa City, IA, USA. [5] Laboratory of Genetic Instability and Oncogenesis, UMR8200 CNRS, Gustave Roussy, Université Paris-Sud, Villejuif, France. [6] Department of Pathobiology, Faculty of Science, Mahidol University, Bangkok, Thailand. [7] Cancer Epigenetics Program, Fox Chase Cancer Center, Philadelphia, PA, USA. [8] Ludwig Institute for Cancer Research, Nuffield Department of Clinical Medicine, University of Oxford, Headington, Oxford, UK. [9] Department of Oncology, Clinical Sciences Lund, Lund University and Skåne University Hospital, Lund, Sweden. [10] Centre of Neuroregeneration, University of Edinburgh, Edinburgh, UK. [11] Department of Biochemistry and Molecular Biology, and Department of Anatomy, BioMedical Center, Faculty of Medicine, University of Iceland, Reykjavik, Iceland. [12] Department of Functional Genomics and Cancer, Institut de Génétique et de Biologie Moléculaire et Cellulaire, CNRS/INSERM/UNISTRA, 1 Rue Laurent Fries, 67404 Illkirch, Cedex, France. [13] These authors contributed equally: M Hamm, P Sohier, V Petit. ✉email: colin.goding@ludwig.ox.ac.uk; lionel.larue@curie.fr

Cancer initiation is triggered by the activation of oncogenic signaling combined with senescence bypass. Yet while many typical oncogenes and tumor suppressors that affect cancer initiation have been identified, cancer initiation is likely to be modulated by additional genetic events. Understanding how non-classical driver mutations may impact cancer initiation is a key issue that has been relatively underexplored. Melanoma, a highly aggressive skin cancer, arises through the acquisition of well-defined genetic and epigenetic modifications in oncogenes and tumor suppressors and represents an excellent model system to address this key question.

As a highly genetically unstable cancer type, the initiation of melanoma requires the induction of melanocyte proliferation, which is mediated by several major founder mutations, the most common of which are $BRAF^{V600E}$ and $NRAS^{Q61K/R}$[1,2]. However, activation of BRAF or NRAS is insufficient to promote melanoma initiation without senescence bypass mediated by additional founder mutations or expression changes of several genes including $p16^{INK4A}$, CTNNB1, PTEN, or MDM4[3–7].

The transcription factor BRN2, also known as POU3F2 and N-OCT3, plays a critical role in neurogenesis and drives proliferation in a range of cancer types with neural or neuroendocrine origins, including glioblastoma, neuroblastoma, small cell lung cancer, and neuroendocrine prostate cancer[8–10]. In the melanocyte lineage, BRN2 is not detected in melanoblasts in vivo but is heterogeneously expressed in naevi and melanoma[11–14]. In vitro studies have shown that BRN2 expression is induced by a range of melanoma-associated signaling pathways including activation of the mitogen-activated protein kinase (MAPK) pathway downstream from BRAF, the PI3K pathway, the LEF-β-catenin axis, as well as FGF, TNF-α, EDN3, and SCF signaling[14–17]. Consistent with BRN2 being expressed in a predominantly mutually exclusive pattern with the Microphthalmia-associated transcription factor (MITF)[13] that plays a crucial role in melanoma proliferation[18], BRN2 is repressed by MITF via miR-211[19]. However, the relationship between MITF and BRN2 is complex. For example, BRN2 was recently found to be regulated by E2F1, a cell cycle-regulated transcription factor that is also a target for MITF, and both BRN2 and MITF are regulated by PAX3 and WNT/β-catenin[15,16,20–24]. Indeed, both BRN2 and MITF can regulate expression of AXL[21,25], with BRN2 repressing AXL expression, thus enabling some cells in human melanoma to adopt an AXL$^{High}$, MITF$^{Low}$ and BRN2$^{Low}$ state[25]. The activation of BRN2 expression in a specific subset of melanoma cells in response to all three major signaling pathways (MAPK, PI3K/PTEN, and β-catenin) linked to melanoma initiation (early proliferation and bypass/escape senescence) and progression (including late proliferation and metastatic dissemination) suggests that Brn2 is likely to have a critical role in disease progression[26]. Most notably, BRN2 has been associated with MITF$^{Low}$ senescent or slow-cycling cells[11], and identified as a key regulator of melanoma invasion and anoikis in vitro[13,27,28] and in in vivo xenograft experiments[20,29,30]. Mechanistically, the ability of BRN2 to promote invasion has been linked to its ability to control expression of PDE5A-mediated cell contractility, phosphorylation of myosin light chain 2, repression of MITF and PAX3, and cooperation with bi-allelic loss of CDKN2A[13,20,27,30]. However, despite abundant information linking BRN2 to melanoma proliferation and invasiveness in vitro and in xenograft experiments, the impact of BRN2 on melanoma initiation and progression in vivo has never been assessed.

In this work, we show that BRN2 acts as a tumor suppressor during melanoma initiation and progression in a BRAF-PTEN context since BRN2 and MITF regulate positively and negatively the transcription of PTEN, respectively.

## Results

**BRN2 loss or low expression correlates with reduced survival and worse prognosis.** Although BRN2 has been implicated in melanoma invasiveness, and its expression is highly regulated, whether and how it may contribute to melanoma initiation or incidence is not understood. To evaluate the prevalence of BRN2 loss in human skin cutaneous melanoma (SKCM), we retrieved copy-number alteration (CNA) data for BRN2 in SKCM metastases (stage IV) from the Cancer Genome Atlas (TCGA, https://cancergenome.nih.gov/). The BRN2 locus showed mono-allelic loss in 53% and bi-allelic loss in 2.7% of all patient samples ($n = 367$, Fig. 1A). Only a minority ($n = 29$ of 367, corresponding to 7.9%) of SKCM samples showed a copy-number gain ($n = 27$)/amplification ($n = 2$) for BRN2. We did not further analyze these samples since the expression of BRN2 was slightly increased but not statistically different ($p = 0.26$, test of Kruskall-Wallis with a Dunn correction) between tumors that gained and/or amplified this locus compared to the normal situation. We screened a panel of human melanoma cell lines available in our laboratory ($n = 23$) for deletions that affect the BRN2 locus by comparative genomic hybridization. The BRN2 locus showed mono-allelic loss in 48% (11 out 23) of the human melanoma cell lines and no bi-allelic loss, comparable to the TCGA-data (Supplementary Fig. 1A, Supplementary Data 1). Notably, BRN2 mRNA levels were significantly lower in SKCM metastases with bi-allelic BRN2 loss (Supplementary Fig. 1B). The mono- and bi-allelic loss of BRN2 was frequently associated with a large segmental deletion of the long arm of chromosome 6 (Chr.6q) in SKCM metastases and in our cell line panel (Fig. 1B, Supplementary Fig. 1C, Supplementary Data 2). From the TCGA, patients carrying the monoallelic loss of BRN2 in metastases displayed a trend to have a shorter overall survival than those with diploid status (Fig. 1C). These results were validated using an independent cohort of 108 regional metastases previously described[31] (Fig. 1D). Moreover, we evaluated the number of BRN2 alleles in nevi and melanoma that arose from these nevi using publically available data[32]. It appears that 28% (5 out of 18) or 22% (4 out of 18) of melanomas presented either a mono-allelic loss or a gain of BRN2 respectively compared to nevi (Supplementary Fig. 1D). The situation is clearly complex, but we may conclude that BRN2 mono-allelic loss can occur during the early steps of melanomagenesis.

We next assessed the correlation between BRN2 mRNA levels and overall patient survival to evaluate the effect of BRN2 mono-allelic loss on melanoma progression. We established "BRN2-high" and "BRN2-low" patient groups based on RNA-seq data available from the TCGA (BRN2 subgroups defined as BRN2-low (≤1 transcript per million reads [TPM]) and BRN2 expressed/high [>1 TPM]). Patients in the "BRN2-low" group displayed significantly shorter overall survival than those of the "BRN2-high" group (Fig. 1E). Overall, the BRN2 locus, frequently associated with a large segmental deletion, is lost (mono- and bi-allelic) in ≈60% of human SKCM metastases and correlates with significantly reduced overall survival. Since CCNC, ROS1 and ARID1B loci are distal to BRN2 on chromosome 6, and are known to be involved in melanomagenesis, we evaluated the overall survival of these patients according to the presence and the level of mRNA expression of the corresponding genes. We observed no significant difference between the presence or absence of these three genes or their expression (Supplementary Fig. 1E–L). Finally, we compared the loss of 6q with the mono-allelic loss (MAL) of BRN2 (Supplementary Fig. 1M) and found that 6q loss is associated with a worse prognostic than BRN2 mono-allelic loss. This result indicates, as suspected, that other gene(s) located on 6q are of importance in melanomagenesis. Taken together, these data indicate that in human melanoma

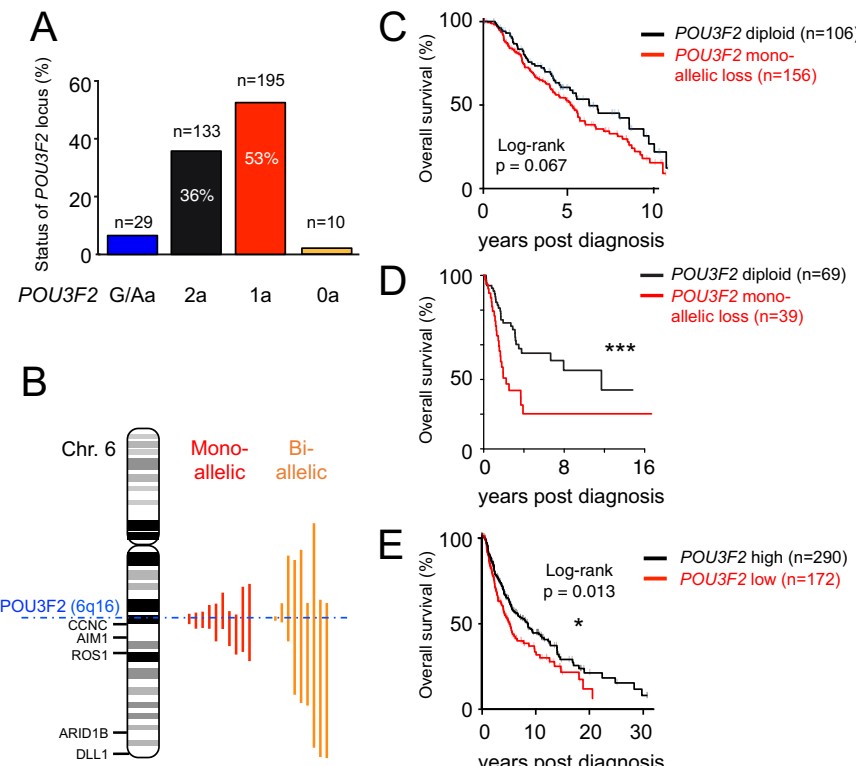

**Fig. 1 One *BRN2* allele is frequently lost in human melanoma and reduced BRN2 mRNA level correlates with reduced overall survival. A** Bar graph showing the status of the *BRN2* locus in human skin cutaneous melanoma (SKCM) metastases (stage IV). Copy-number alterations (CNAs) were estimated using the GISTIC algorithm. Two alleles (2a in black), one allele (1a in red), no allele (0a in orange), and gain and/or amplification (G/Aa in blue) of the *BRN2* locus are given. **B** Pictogram showing the extent of segmental deletions (red or orange vertical lines) that affect the *BRN2* locus on Chr.6q16 (dashed blue horizontal line) in SKCM metastases. **C** Kaplan–Meier curves comparing 10-year overall survival of SKCM patients diploid for BRN2 (black line, $n = 106$) or those with mono-allelic (red $n = 156$). The TCGA CNA-data set was analyzed ($n = 309$). Diploid vs. Mono-allelic loss: log-rank (Mantel-cox) test ($p = 0.067$). Data were retrieved from TCGA on August 8, 2019. **D** Kaplan–Meier curves comparing melanoma patients with diploid status or mono-allelic loss of *BRN2* in 108 regional metastatic melanoma patients ($p = 0.001$, log-rank test)[31] and unpublished data. **E** Kaplan–Meier curves comparing 30-year overall survival of SKCM patients to BRN2 mRNA levels. Log-rank (Mantel-Cox) test ($p = 0.03$). Data were retrieved from TCGA on August 8, 2019. Significance was defined as * ($p < 0.05$) and *** ($p \leq 0.001$).

BRN2 loss/low expression is associated with an adverse outcome for the patient.

**Co-occurrence of *BRN2* loss with mono-allelic loss of *PTEN*.** We next determined whether BRN2 loss co-occurs with melanoma driver mutations by examining the TCGA CNA-data set and human melanoma cell-line panel. There was no significant correlation between *BRAF* or *NRAS* mutation and *BRN2* loss (mono- or bi-allelic), neither in human melanoma samples nor the human cell-line panel (Supplementary Fig. 2A,B). We then searched for co-occurring CNAs of other known melanoma-associated genes and found that mono-allelic loss of *BRN2* co-occurred with mono-allelic loss of *PTEN* in approximately 40% of the human melanoma samples in TCGA and in the human cell-line panel (Supplementary Fig. 2C,D). We next evaluated the concomitant *BRN2* locus alterations, *BRAF/NRAS* mutations, and *CDKN2A/PTEN* alterations and found the most frequent genetic constellation that co-occurs with *BRN2* loss in melanoma to be *BRAF*[V600X] mutation together with mono-allelic *PTEN* loss (Supplementary Fig. 2E). Finally, we compared the overall survival of human patients with a loss of one allele of *PTEN* who also had a loss of *BRN2* (monoallelic loss = MAL) with a loss of one allele of *PTEN* and no loss of *BRN2* (BRN2-normal). In this context, patients with loss of *BRN2* showed significantly lower overall survival than *BRN2*-normal patients (Supplementary Fig. 1N).

In conclusion, in human melanoma, the loss of *BRN2* is preferentially associated with *BRAF* mutation together with *PTEN* loss.

**Loss of *BRN2* drives melanomagenesis in vivo**. These data suggesting that the loss of *BRN2* might be of importance in melanoma prompted us to evaluate the potential causal role of BRN2 in melanomagenesis in vivo by examining whether heterozygous (het) or homozygous (hom) loss of *Brn2* affects melanoma initiation and/or progression in a mouse model. Note however, that while genetic loss of BRN2 might be important, the complex regulation of BRN2 expression driven by melanoma-associated signaling pathways might also play a major role, especially given that melanoma cells within a single tumor can exhibit both high and very low BRN2 expression[13,20]. We therefore developed an inducible genetically engineered mouse model system for generating Brn2-deficient melanoma driven by the most common alterations in human SKCM (*Braf*[V600E] and *Pten* loss). Specifically, we used *Tyr::Cre*[ERt2/°-Lar]; *Braf*[V600E/+] (called Braf from hereon) and *Tyr::Cre*[ERt2/°-Lar]; *Braf*[V600E/+]; *Pten*[F/+] (called Braf-Pten from hereon) mice carrying a tamoxifen-inducible Cre-recombinase under the control of the tyrosinase promoter[33–35]. This model system allows melanocyte lineage-specific induction of a *BRAF*[V600E] mutation and mono-allelic deletion of *Pten* for Braf-Pten mice. Cre-mediated defloxing leads to activation of the Braf[V600E] oncogene, inducing nevus

and spontaneous melanoma formation in Braf mice, reproducing many of the cardinal histological and molecular features of human melanoma[36]. Bi-allelic and mono-allelic loss of *PTEN* reduces tumor latency in Braf$^{V600E}$- and NRAS$^{Q61K}$-driven mouse melanoma models[3,37].

Using these models, we studied the effect of Brn2 insufficiency (het and hom) on in vivo melanomagenesis by introducing the floxed Brn2 locus into the genome by appropriate crossings (Supplementary Fig. 3A)[38]. Specifically, we generated the mouse lines *Tyr::Cre$^{ERT2/°}$; Braf$^{V600E/+}$; Brn2$^{+/+}$* (Braf-Brn2-WT), *Tyr:: Cre$^{ERT2/°}$; Braf$^{V600E/+}$; Brn2$^{F/+}$* (Braf-Brn2-het), *Tyr::Cre$^{ERT2/°}$; Braf$^{V600E/+}$; Brn2$^{F/F}$* (Braf-Brn2-hom), *Tyr::Cre$^{ERt2/°}$; Braf$^{V600E/+}$; Pten$^{F/+}$; Brn2$^{+/+}$* (Braf-Pten-Brn2-WT), *Tyr::Cre$^{ERt2/°}$; Braf$^{V600E/+}$; Pten$^{F/+}$; Brn2$^{F/+}$* (Braf-Pten-Brn2-het) and *Tyr::Cre$^{ERt2/°}$; Braf$^{V600E/+}$; Pten$^{F/+}$; Brn2$^{F/F}$* (Braf-Pten-Brn2-hom). Cre-mediated defloxing of *Braf*, *Pten*, and *Brn2* loci was induced by topical application of tamoxifen during the first 3 days after birth (Supplementary Fig. 3B). All mice were monitored for the appearance and growth rate of the first tumor, as well as for the number of tumors/mouse. Note that the ability to generate either homo- or heterozygous Brn2 KOs will mimic not only mono or biallelic loss in humans, but also reflect the variable levels of BRN2 observed within human tumors[13,20]. In the absence of PTEN (*Pten$^{F/F}$*) on a Braf$^{V600E}$ background, the appearance of the tumors was too rapid to observe any difference between Brn2 + /+, Brn2 F/ + or Brn2 F/F mice.

Braf-Brn2-WT/het/hom mice showed no differences in the appearance of the first tumor, number of tumors/mouse or the tumor growth rate from those of Braf-WT mice (Fig. 2, Supplementary Fig. 3C). However, compared to Braf-Pten-Brn2-WT, Braf-Pten-Brn2-het/hom mice significantly increased the number of tumors/mouse and the tumor growth rate, but not the timing of the appearance of the first tumor (Fig. 2, Supplementary Fig. 3D). We verified that *Brn2*, *Braf*, and *Pten* were correctly defloxed in the resulting melanomas (Supplementary Fig. 3E,F). In summary, our data show that Brn2 acts as a tumor suppressor in vivo. As it has been shown in humans and mice, induction of early proliferation induced by the presence of the BRAF$^{V600E}$ mutation leads to senescence, but it can be bypassed when the level of Pten is reduced[3,37,39,40]. It is important to note that in the absence of Pten in the physiological context, proliferation of melanoblasts and melanocytes is not induced as the reduction/lack of Pten promotes proliferation once the cells are transformed[3,41]. On a Braf-Pten context the loss/reduction of Brn2 appears to induce melanoma initiation after promoting proliferation and bypass/ escape of senescence, and then allows the tumor growth as was previously showed in vivo[3,37].

**Reduction of BRN2 levels increases proliferation in vivo and in vitro.** The effect of *Brn2* loss on tumor growth prompted us to investigate whether *Brn2* loss increases intra-tumor proliferation. Staining sections for Ki-67, a marker of cycling cells, revealed that melanomas from Braf-Pten-Brn2-het/hom mice displayed a significantly higher number of Ki-67$^+$ cells than Braf-Pten-Brn2-WT melanomas (Fig. 3A,B). To confirm this result, we injected Braf-Pten-Brn2 mice with bromodeoxyuridine (BrdU) two hours prior to euthanization, to determine whether melanoma cells were slow or fast-dividing. Braf-Pten-Brn2 melanomas had a significantly higher number of BrdU$^+$ cells when Brn2 was heterozygous or homozygous (Fig. 3C,D). These results indicated that heterozygous loss of *Pten* combined with heterozygous/homozygous loss of *Brn2* promotes melanoma proliferation in vivo.

We next assessed whether Brn2 knockdown favors proliferation in vitro and whether this mechanism is conserved (i) between human and mouse and (ii) between transformed and

non-transformed cells using Dauv-1 human melanoma cell line, and the Melan-a mouse melanocyte cell line. These cell lines express both Pten and Brn2 mRNA and protein (Supplementary Table 1). Dauv-1 cells carry a BRAF$^{V600E/+}$ mutation identical to that used in the mouse melanoma model system, and Melan-a cells are WT for Braf. siRNA-mediated knockdown of Brn2 significantly increased cell number 72 h after transfection of both cell lines (Fig. 3E–G). Brn2 knockdown, assessed by western blot, led to an increase in cyclin D1 protein levels in both cell lines, but did not alter cyclin D1 mRNA levels, suggesting a regulation of cyclin D1 at the protein level (Fig. 3F–H). Overall, the reduction of the Brn2 protein induces cell proliferation of the melanocytic lineage in vivo and in vitro.

Colony formation assays in vitro are classically used to show the importance of a gene in tumorigenesis, and indicate that a single cell may survive in vitro and proliferate to form a colony. Previous work has demonstrated that a reduction in Brn2 levels in melanoma cells has no effect on colony formation[42,43]. However, to test this in our model, mouse melanoma cell lines were established and characterized from the different independent Braf-Pten-Brn2 C57BL/6 J mouse melanoma; the m50 and m6 cell lines were Brn2-WT, m59, and m36 were Brn2-het, and m82 and m8 were Brn2-hom (Supplementary Fig. 4A–F). The presence or absence of Brn2 did not decrease the ability of these melanoma cell lines to grow in syngeneic mice (Supplementary Table 2). In other words, it appears that the absence of Brn2 in these melanoma cells does not affect the implantation of the cells on the body wall, the proliferation after their transformation or the induction of angiogenesis in an immunocompetent environment. We evaluated the capacity of the m50 (Brn2 WT), m59 (het), and m82 (hom) cells to generate colonies and observed, in agreement with previous observations obtained using BRN2 depletion, that the three cell lines have similar abilities to form colonies (Supplementary Fig. 4G). Moreover, re-expression of Brn2 using lentivirus infection in two independent m8 and m82 Braf-Pten-Brn2-hom mouse melanoma cell lines, also indicated that BRN2 does not affect their capacity to form colonies under these conditions (Supplementary Fig. 4H, I). In conclusion, as shown in human cell lines, the number of colony forming units is independent of the presence or absence of Brn2 and consequently this assay is uninformative regarding the role of Brn2 in melanomagenesis.

We also evaluated the activity of BRAF$^{V600}$ (PLX4720), MEK (Binimetinib), and PI3K (LY294002) inhibitors on the capacities of m50, m59 and m82 mouse melanoma cell lines to form CFU and determined the IC50 of these drugs (Supplementary Fig. 4J-O). Consistent with observations suggesting that BRN2 may suppress cell death[43], Brn2-het/hom cells are more sensitive than WT cells to these three drugs. We also evaluated the cooperativity of PLX4720 and LY294002 using various concentrations of each drug, but we did not observe any cooperation/synergy of these two drugs.

**Mono-allelic loss of Brn2 induces melanoma metastasis.** We next evaluated the effects of Brn2 on metastasis formation in vivo. Since human SKCM is known to spread to proximal lymph nodes (LNs), we assessed the presence of pigmented cells in the inguinal LNs of tumor-bearing Braf-Pten-Brn2 mice. Specifically, we estimated the volume of the various metastases present in LNs and the number of pigmented areas after haematoxylin & eosin (HE) staining of LN sections (Fig. 4A–C). All Braf-Pten-Brn2 mice, irrespective of BRN2 status, showed the presence of pigmented cells in both inguinal LNs. However, the LN volume of Braf-Pten-Brn2-het mice was significantly higher than that of Braf-Pten-Brn2-WT and Braf-Pten-Brn2-hom (Fig. 4A, B). Similarly, Braf-

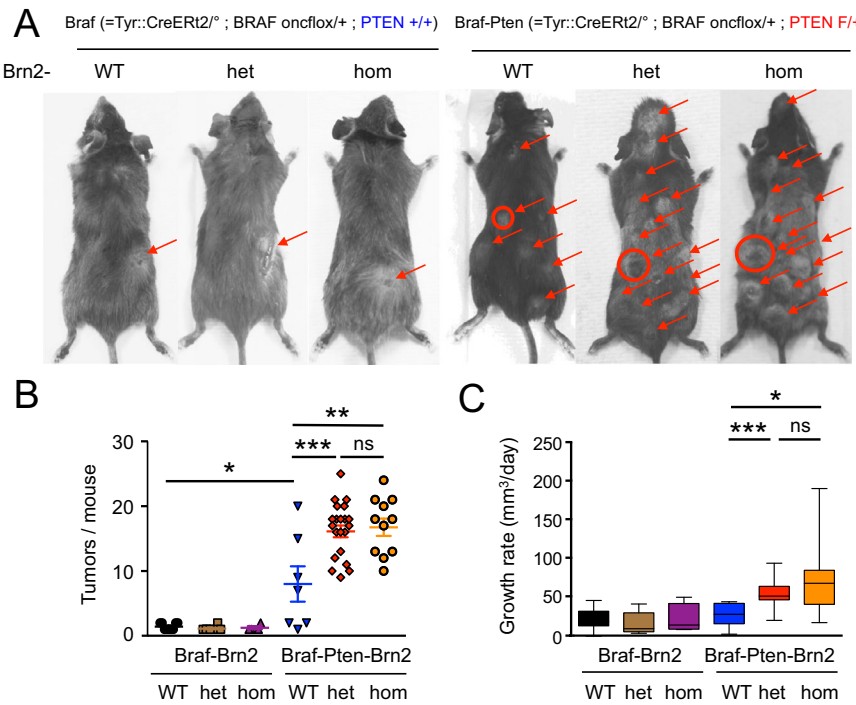

**Fig. 2 Brn2 loss potentiates melanomagenesis in Braf-Pten mice. A** Macroscopic pictures of the dorsal view of mice with cutaneous melanomas carrying mutations in the melanocyte lineage for Braf, Pten, and Brn2 after tamoxifen induction at birth (p1, p2, and p3 – see Supplementary Fig. 3). *Tyr::CreER*$^{T2/°}$; *Braf*$^{V600E/+}$; *Pten*$^{+/+}$ (= Braf), *Pten*$^{F/+}$ (= Pten), *Brn2*$^{+/+}$ (=Brn2-WT), *Brn2*$^{F/+}$ (= Brn2-het), and *Brn2*$^{F/F}$ ( = Brn2-hom). Tumors are highlighted with arrows and the sizes of the first growing tumors to appear are proportional to the diameters of the circles. F means floxed allele. **B** All Braf ($n = 9$), Braf-Brn2-het ($n = 8$), and Braf-Brn2-hom ($n = 4$) mice produced cutaneous melanomas and their number was similar (1 to 2 tumors/mouse). All Braf-Pten-Brn2-WT ($n = 7$), Braf-Pten-Brn2-het ($n = 21$), and Braf-Pten-Brn2-hom ($n = 11$) mice produced cutaneous melanomas. Note that in the absence of Pten (*Pten*$^{F/F}$), the appearance of the melanoma was too rapid to observe any difference between Brn2-WT, Brn2-het, and Brn2-hom mice. Each dot corresponds to an individual mouse. As control, mice of different genetic backgrounds were produced and not induced with tamoxifen Braf [$n = 12$], Braf-Brn2-het [$n = 25$], Braf-Brn2-hom [$n = 11$], Braf-Pten [$n = 7$], Braf-Pten-Brn2-het [$n = 13$], and Braf-Pten-Brn2-hom [$n = 6$]; none of them developed melanoma after 18 months, except one Braf-Pten-Brn2-het mouse that developed one melanoma after 12 months. None of the mice that were wild-type for *Braf* displayed any obvious phenotype, irrespective of the status of *Pten* or *Brn2*, including melanomagenesis and hyperpigmentation. **C** Growth rates of the first tumor appearing in each mouse for Braf-Brn2-WT, Braf-Brn2-het, Braf-Brn2-hom, Braf-Pten-Brn2-WT, Braf-Pten-Brn2-het, and Braf-Pten-het-Brn2-hom mice. The number of tumors is determined all along the life of the mouse by checking the mice a minimum of twice a week. Statistical analysis was performed using the two-tailed unpaired *t* test. ns = non-significant, *$p < 0.05$, **$p < 0.01$, and ***$p < 0.001$. Data are presented as mean values ± SEM. Braf-Pten-Brn2-het mice were euthanized in average 1.3 weeks after appearance of the first tumors with an average of 16 tumors/mouse. Similar results were obtained with Braf-Pten-Brn2-hom mice. Braf-Pten-Brn2-WT mice were euthanized at 4 weeks with an average of 8 tumors even though they did not reach a total volume of 2 cm$^3$ except for one mouse that was euthanized earlier (3 weeks).

Pten-Brn2-het LNs showed a higher number of pigmented areas per mm$^2$ than Braf-Pten-Brn2-WT/hom mice (Fig. 4C).

To verify that the pigmented cells in the lymph nodes did not arise from cells in which the Cre recombinase had not worked efficiently, we tested whether these pigmented cells were properly defloxed for Brn2. The targeted Brn2-flox allele, used in our mouse model, has an eGFP-cassette inserted downstream of the floxed *Brn2* locus (Fig. 4D). Thus the production of eGFP occurs once the upstream *Brn2* gene is defloxed. Consistent with correct defloxing of *Brn2*, pigmented areas of LNs expressed GFP in Braf-Pten-Brn2-het/hom mice, but not Braf-Pten-Brn2-WT mice (Fig. 4E). The pigmented cells present in Braf-Pten-Brn2-WT and Braf-Pten-Brn2-het LNs also expressed Sox10, a melanocytic marker (Fig. 4E), that co-localized with eGFP-expression in Braf-Pten-Brn2-het mice confirming the melanocytic origin of the pigmented cells observed.

To get a better understanding of melanomagenesis in the various Brn2-WT/het/hom situations, we performed a transcriptomic analysis of the various Brn2 tumors and cell lines (Supplementary Data 3 and 4). The ontology enrichment analysis indicated that the Braf-Pten-Brn2-het tumors were more inflamed than the Braf-Pten-Brn2-WT tumors with increased neutrophil-

associated gene expression (Supplementary Fig. 5A). It also suggests that the extracellular-matrix was actively remodeled and that these Braf-Pten-Brn2-het tumors were more subject to angiogenesis (Supplementary Fig. 5A). Pathway enrichment analysis supported these results as inflammatory gene expression signatures are enriched from both the WikiPathways and KEGG databases (Supplementary Fig. 5B,C). Significantly, the PI3K/AKT pathway was enriched in the Braf-Pten-Brn2-het tumors compared to the Braf-Pten-Brn2-WT in both WikiPathways and KEGG databases (Supplementary Fig. 5B,C), suggesting that AKT could be more phosphorylated in the Braf-Pten-Brn2-het tumors. This assumption was verified by Western-blot analysis where phosphorylation of AKT on S473, a surrogate of the AKT activity, and the phosphorylation of S6 on S235/236 were significantly increased in Braf-Pten-Brn2-het/hom tumors compared to Braf-Pten-Brn2-WT tumors (Supplementary Fig. 5D).

Gene Set Enrichment Analysis (GSEA) using the melanoma invasive signature from Verfaillie[44] indicated that Braf-Pten-Brn2-het tumors and cell lines were more invasive than Braf-Pten-Brn2-hom tumors and cell lines (Fig. 4F). Moreover, from the GO, WikiPathways and KEGG 2019 analyses of the mouse melanoma tumors, the immune system (cytokine, neutrophil,

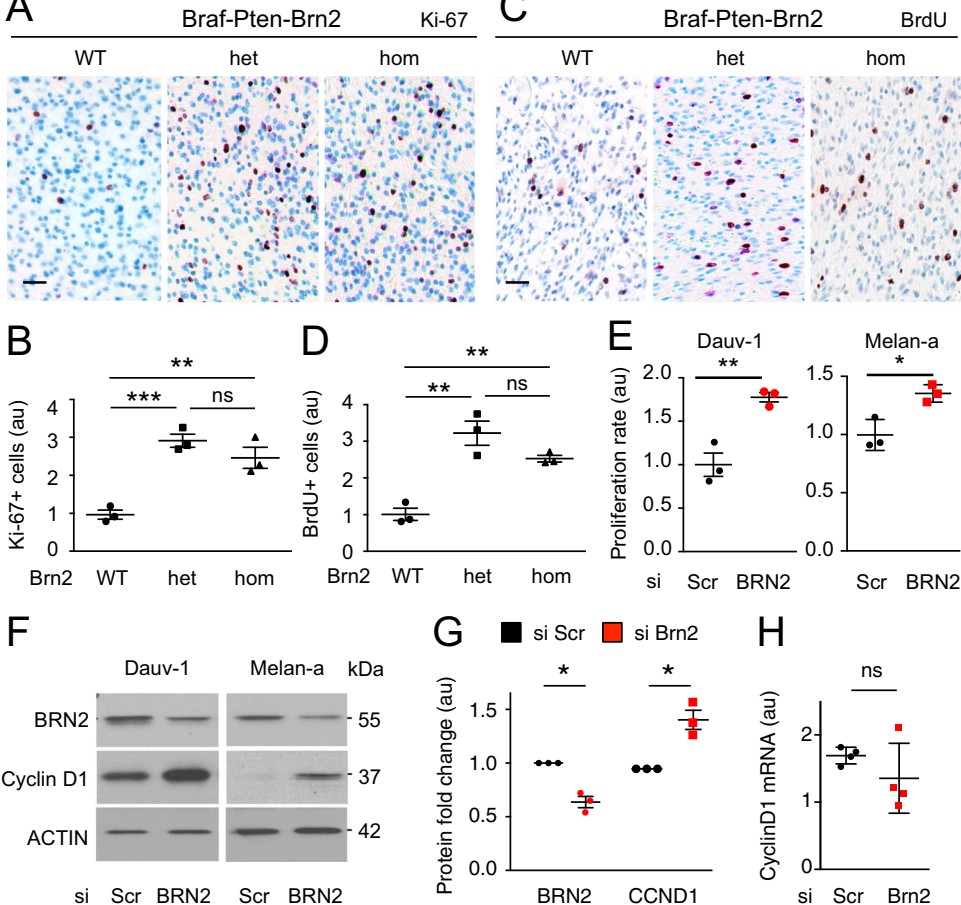

**Fig. 3 BRN2-het/hom induces proliferation in vitro and in vivo. A–D** Representative photomicrographs of Ki-67 (**a**) and BrdU (**c**) stainings of Braf-Pten-Brn2-WT/het/hom tumors. Ki-67+ cells are stained in red. Nuclei are stained in blue. Scale bar = 40 μm. Quantification of (n = 3) Ki-67+ (**B**) and BrdU (**D**) stainings of (**A**) and (**C**), respectively. Scale bar = 40 μm. Each dot represents the result for one tumor. (**e**) Growth rate is induced in Dauv-1 and Melan-a cell lines after reduction of Brn2 using siBRN2 and siScr as control (Scr = scramble). Three independent biological and technical experiments were performed for each cell line and for each condition. **F–H** Brn2 knock down induces Cyclin D1 protein but not its mRNA in melanocytes. **F** Western blot analysis for Brn2, Cyclin D1, and actin after reduction of Brn2 in Dauv-1 and Melan-a cells. Experiments were performed independently three times. One representative western blot is shown (raw data are presented in Supplementary Fig. 8). Quantification of protein (**G**) and mRNA (**H**) levels for Dauv-1 cells after siRNA-mediated knockdown (n = 3, independent experiments). For the proteins, all values were normalized against the background and corresponding actin loading control for each sample. Quantification was performed using *Image-J* software. For mRNA, all values were normalized against those of TBP. au = arbitrary units. Statistical analysis was performed using the two-tailed unpaired (**B, D, E, H**) and paired (**G**) *t* tests. ns = non-significant, *p < 0.05, **p < 0.01, and ***p < 0.001. Data are presented as mean values ± SEM.

macrophage) and angiogenesis are induced in Braf-Pten-Brn2-het tumors compared to Braf-Pten-Brn2-WT tumors (Supplementary Fig. 5A–C). This information suggests that Braf-Pten-Brn2-het cells have more potential to metastasize than Braf-Pten-Brn2-WT and Braf-Pten-Brn2-hom cells. In this respect, we tested the capacity of Braf-Pten-Brn2-hom (m82) mouse melanoma cell lines re-expressing or not Brn2 (m82 and m82 + Brn2) to invade matrigel in 3D. In the presence of ectopic Brn2, the m82 Brn2 KO melanoma cell lines have more ability to invade (Fig. 4G and Supplementary Fig. 4I). Since AXL, a receptor tyrosine kinase (RTK), is associated with melanoma metastasis[45], we also evaluated the level of Axl in mouse melanoma tumors and cells. Braf-Pten-Brn2-het cells produce slightly more Axl mRNA than Brn2-WT/hom (Fig. 4H). This upregulation does not affect genes that are in *Cis* of Axl suggesting a specific regulation of Axl in a Braf-Pten-Brn2-het context (Supplementary Fig. 5E–G). Moreover, when the levels of both BRN2 and MITF are reduced in human melanoma cell lines, the level of AXL mRNA is induced (Supplementary Fig. 6D,H,L,P,T). More precisely the level of AXL is slightly, but significantly, higher in Gerlach and DAUV-1 cell

lines in which the level of AXL is already very high, and is higher in SK28 and 501Mel cell lines in which the level of AXL is much lower than in Gerlach and DAUV-1 cell lines.

In conclusion, compared with Braf-Pten-Brn2-WT mice, melanoma initiation is promoted in both Braf-Pten-Brn2-het/hom mice, but metastasis is promoted only in Braf-Pten-Brn2-het mice. These observations are consistent with the fact that on a Brn2-het/hom background melanoma initiation (proliferation and bypass/escape of senescence) is induced and melanoma invasion and/or survival is inhibited on a Brn2-hom background.

**BRN2 binds to the *PTEN* promoter and BRN2 loss leads to the reduction of *PTEN* transcription.** The PI3K-AKT pathway is induced in melanoma and its induction abrogates BRAF$^{V600E}$-induced senescence[46,47]. The loss of *Pten*, a suppressor of the PI3K pathway, induces melanoma initiation and proliferation in vivo[3,37]. Since our Braf-Pten mouse melanoma model retained one functional allele of *Pten*, we hypothesized that *Brn2* loss would induce less expression from the WT *Pten* allele, leading to

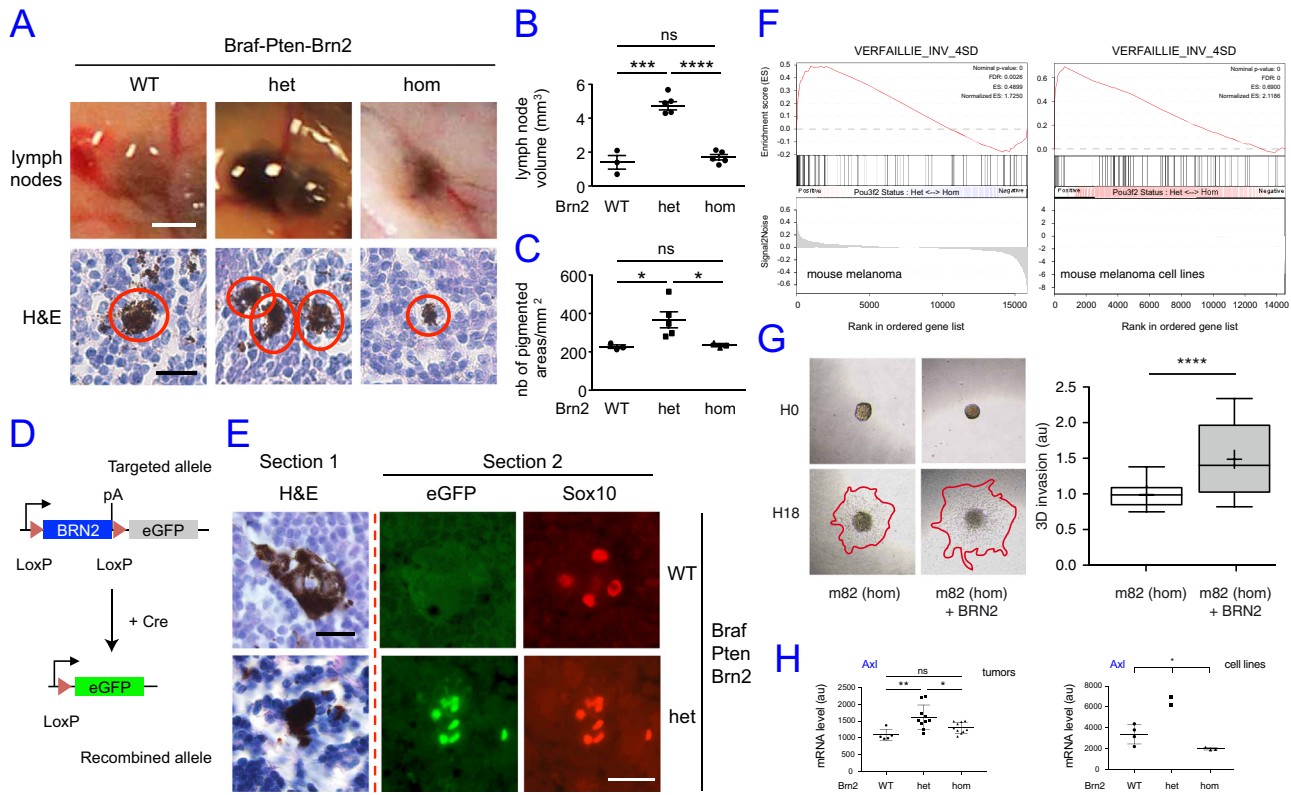

**Fig. 4 Mono-allelic loss of *Brn2* induces melanoma metastasis. A** Upper panel: Representative photomicrographs of in situ inguinal lymph nodes (LN) of Braf-Pten-Brn2-WT/het/hom mice. Scale bar = 1 mm. The pigmented volume (mm³) was estimated for each LN. Lower panel: Representative photomicrographs of haematoxylin & eosin (H&E) staining of LNs containing pigmented cells. Scale bar = 20 µm. **B** Quantification of the pigmented volume of inguinal LNs in the upper panel of Figure (**A**). $n = 3, 5$, and 5 for WT, het, and hom. (**c**) Quantification of the pigmented areas per mm² of inguinal LNs in the lower panel of Figure (**A**). Pigmented areas > 50 µm² were considered. $n = 3, 5$, and 3 for WT, het, and hom. **D** Scheme showing the defloxing strategy of Brn2 in melanocytes of the primary tumor that releases eGFP expression upon the defloxing of *Brn2*. **E** Representative photomicrographs of serial LN sections of Braf-Pten-Brn2-WT and Braf-Pten-Brn2-het mice stained with H&E and the melanocyte marker Sox10. H&E staining was evaluated for one section and GFP (green channel) and Sox10 staining (red channel) evaluated for an adjacent section. Scale bar = 20 µm. **F** A melanoma invasive signature was significantly enriched in the Braf-Pten-Brn2-het tumors (left) and in the Braf-Pten-Brn2-het melanoma cell lines (right) compared to their Braf-Pten-Brn2-hom counterparts. **G** Left: photomicrographs of m82 and m82-BRN2 mouse melanoma cells embedded as spheroids in 600 µg/mL matrigel at t0 (H0) and 18 hrs after (H18). Right: Boxes and plots represent the area of invasion (red lines on photomicrographs) quantified with *ImageJ* ($n = 54$ for m82 cells and $n = 56$ for m82-BRN2 cells). P value < 0.0001. au = arbitrary unit. **h** Axl mRNA is significantly overexpressed in Braf-Pten-Brn2-het melanoma and melanoma cell lines ($n = 10$ and 2, respectively) compared to the Braf-Pten-Brn2-WT ($n = 5$ and 4, respectively) and Braf-Pten-Brn2-hom ($n = 10$ and 3, respectively) tumors. Statistical analysis was performed using the two-tailed unpaired t test for (**B**), (**C**), (**G**), and (**H**) (tumors) and an Anova test for H (cell lines). Data are presented as mean values ± SEM for (**B**) and (**C**), sd for (**H**), and Box and whiskers (median, min to max) for (**G**). ns = non-significant, *$p < 0.05$, ***$p < 0.001$, ****$p < 0.0001$.

the increased PI3K-AKT signaling observed (Supplementary Fig. 5D) and consequent melanoma initiation and proliferation. We therefore evaluated the number of Pten positive cells in the various Braf-Pten tumors by immunohistochemistry and found that Braf-Pten-Brn2-het/hom tumors showed fewer Pten-pos cells than Braf-Pten-Brn2-WT tumors (Fig. 5A). This result was verified by western blotting of Braf-Pten tumor samples. The reduction of Brn2 correlates with the reduction of Pten in Braf-Pten-Brn2-het tumors compared to the Braf-Pten-Brn2-WT tumors (Fig. 5B). In accordance with the reduced protein levels, the mRNA levels of Brn2 and Pten were significantly lower in Braf-Pten-Brn2-het/hom tumors than in Braf-Pten-Brn2-WT tumors, suggesting regulation of Pten at the transcriptional level (Fig. 5C).

Next, we evaluated the mechanism of Pten repression mediated by reduced levels of Brn2 in the human Dauv-1, Gerlach, SK28, and 501Mel melanoma cell lines and in the non-transformed mouse Melan-a cell line. siRNA-mediated BRN2 knockdown led to significantly reduced PTEN protein and mRNA levels in cell

lines having high BRN2 expression (Fig. 5D, E and Supplementary Fig. 6). In cells with lower level of BRN2 (501Mel), the reduction of PTEN was not observed.

We examined the *PTEN* promoter for BRN2 binding sites conserved between humans and mice to determine whether BRN2 acts directly on *PTEN* and detected two potential Brn2 binding sites (BS1 and BS2) at the positions—2049 and +761 (numbering relative to TSS). Chromatin-immunoprecipitation (ChIP) for BRN2, performed on Dauv-1 melanoma cell extracts, followed by qPCR, revealed quantitative BRN2 binding to both binding sites, comparable to BRN2 binding on the PAX3 promoter (Fig. 5F–H). IgG was used as a negative control.

To evaluate Pten promoter activity in response to BRN2, we cloned a 3.2-kb-human Pten promoter fragment containing the two BRN2 binding sites upstream the luciferase gene to generate the hsPten::Luciferase reporter construct (Fig. 5H). The analysis of human PTEN promoter activity showed an intrinsic activity of hsPten::Luciferase in human melanoma cells (Supplementary Fig. 7A, B). A consistent repression of PTEN transcription was

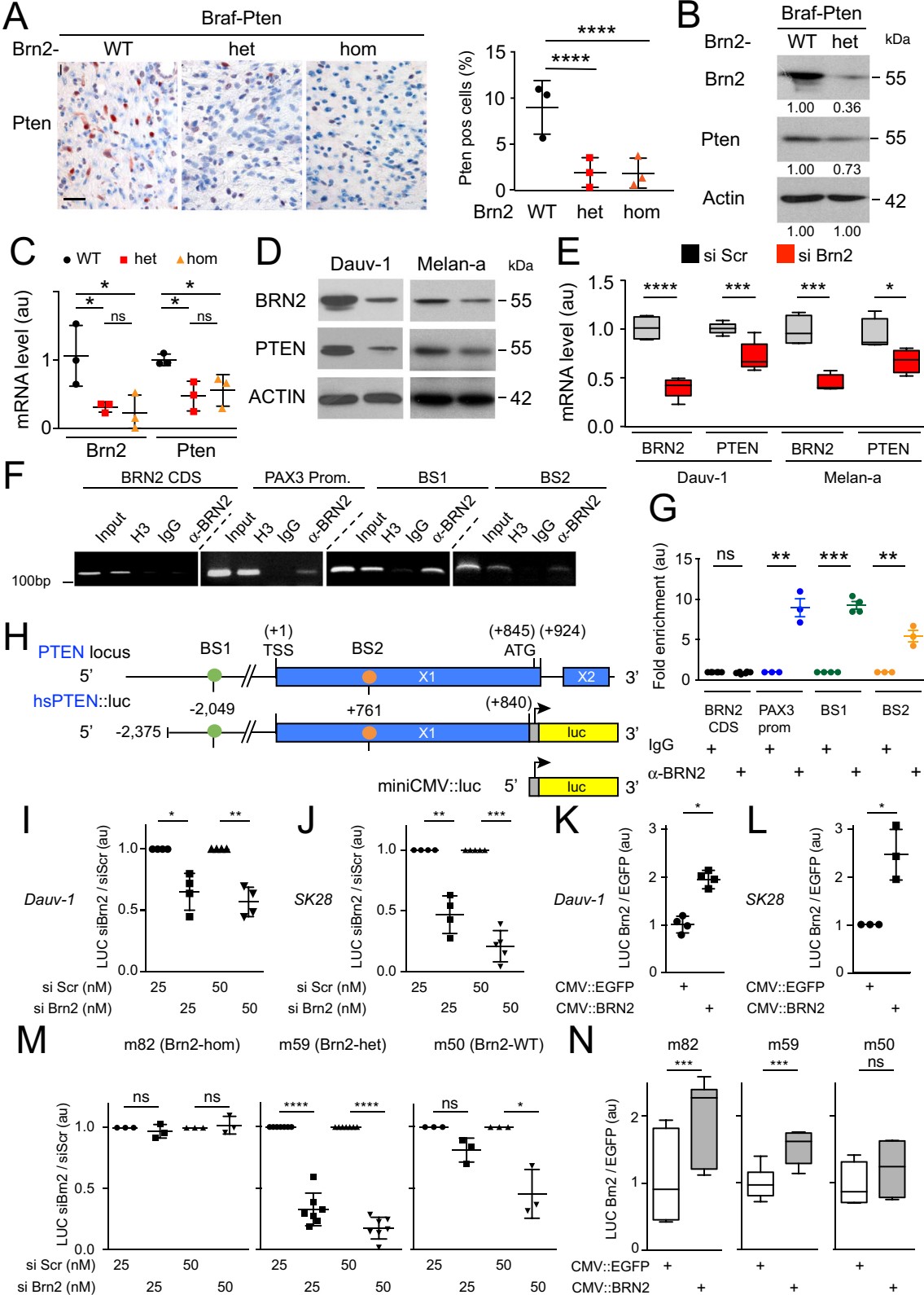

observed when a smart-pool of siBRN2 was added. In contrast, PTEN transcription was activated following co-transfection of a BRN2 expression vector compared to empty vector in Dauv-1 and SK28 cell lines in luciferase assay (Fig. 5I–L). We performed similar experiments with the mouse PTEN promoter, which was cloned upstream of the luciferase construct (Supplementary Fig. 7C–F); Brn2 activated Pten transcription in three mouse melanoma cell lines (m82, m59, and m50) (Fig. 5M, N). In conclusion, BRN2 directly induces PTEN transcription.

**MITF binds the *PTEN* locus and represses *PTEN* transcription.** The *MITF* gene encodes a key transcription factor that plays a major role in melanocyte and melanoma biology[18]. Several

**Fig. 5 Brn2 binds to the Pten promoter and Brn2 loss leads to Pten transcription reduction. A** Representative photomicrographs of immunohistochemistry staining of Pten (red) in Braf-Pten-Brn2-WT and Braf-Pten-Brn2 mouse melanomas are shown. Scale bar = 40 μm. Three independent tumors for each genotype were used for these experiments and three independent sections were used for each tumor. A 2-way ANOVA with Dunnett's multiple comparisons tests were performed. The percentage of Pten$^+$ cells in WT and mutant tumors is shown. **B** Western blot analysis of Brn2, Pten and actin for Braf-Pten-WT and Braf-Pten-Brn2 from at least three tumors of each genotype. One representative example is presented, raw data are presented in Supplementary Fig. 8. The relative intensities of the band were estimated with ImageJ. **C** RT-qPCR of Brn2 and Pten from Braf-Pten-WT and Braf-Pten-Brn2 melanomas. Three independent mouse melanomas per genotype were analyzed. Data were normalized against the values of Gapdh. au = arbitrary unit. **D** Western blot analysis of BRN2, PTEN and ACTIN from Dauv-1 human melanoma cells and Melan-a mouse melanocytes after siRNA mediated knockdown. A representative western blot is shown, raw data are presented in Supplementary Fig. 8. Scr = Scramble. **E** RT-qPCR of BRN2 and PTEN from human melanoma cells (Dauv-1) and mouse melanocytes (Melan-a) after siRNA-mediated knockdown. Specific primers were used for human and mouse samples. Dauv-1 ($n = 6$), Melan-a ($n = 4$), independent experiments. Data were normalized against the values for TBP (Dauv-1) or Gapdh (Melan-a). **F** ChIP assays of BRN2 binding to the PTEN promoter in Dauv-1 melanoma cells. All data shown are representative of at least three independent assays. **G** Quantification of the ChIP-qPCR, plotted and normalized against IgG as the reference. au = arbitrary unit. $n = 6$, 3, 4, and 3 for BRN2 CDS, PAX3 prom, BS1 and BS2, respectively. **H** Scheme of the human PTEN promoter containing two BRN2 binding sites (BS) represented as colored circles. Note that BS are conserved between humans and mice. TSS = transcription start site. Exons (X) 1 and 2 are shown as horizontal rectangles. The translation start site (ATG) and the end of exon 1 are indicated. All numbering is relative to the TSS (+1). Representation of the reporter luciferase (luc) construct with or without PTEN promoter. **I–N** Human and mouse PTEN promoter activities were evaluated in human Dauv-1 (**I**, **K**), SK28 (**J**, **L**), and in mouse m82 [Brn2-hom], m59 [Brn2-het], and m50 [Brn2-WT] (**M**, **N**) melanoma cell lines either in the presence of siScr (scramble) or siBrn2 (smart-pool) (**I**, **J**, **M**) or in the presence of expression vector of BRN2 (CMV::BRN2) (**L**, **N**). The experiments were independently performed four (**I**, **J**, **K**) and three (**L**) times. They were performed three independent times for m82 and m50, seven for m59 (**M**), and seven times for m82, six for m59 and four for m50 (**N**). Statistical analysis was performed using the two-tailed unpaired $t$ test for (**C**), (**E**), (**G**) and paired $t$ test for (**I–N**). Data are presented as mean values ± SEM for (**C**), (**E**), (**G**), **I–M** Box and whiskers (median, min to max) for $N$. ns = non-significant, *$p < 0.05$, ***$p < 0.001$, and ****$p < 0.0001$.

studies have reported that *MITF* transcription is directly repressed by BRN2 whereas a reduction in BRN2 levels leads to increased MITF levels[13,27,48], while another study showed that BRN2 induces MITF[49]. Likely both results are correct due to the versatile role of BRN2 as a transcription regulator whose activity may be dependent on context including genetic status and/or the environment[21,26,27].

Knowing the importance of MITF in the melanocyte lineage, we analyzed the consequences of modulation of MITF expression on PTEN. Using siRNA-mediated *MITF* knockdown in three human melanoma cell lines expressing high levels of MITF (501mel, SK28, HBL) led to a significant increase of PTEN protein and mRNA levels (Fig. 6A, B). In cells expressing lower levels of MITF, such as Gerlach and Dauv-1, we did not observe an increase of PTEN (Supplementary Fig. 6C, G). Next, to determine if MITF directly regulates PTEN expression through proximal enhancers we performed Cleavage Under Targets and Release Using Nuclease (CUT&RUN) with antibodies to MITF, as previously described,[50] and to H3K27Ac, revealing active chromatin, and against H3K4me3, revealing active and poised promoters, in SK28 cell lines that were wild type for *MITF* (MITF-WT) or had loss of function mutations in all alleles of it (ΔMITF) (Fig. 6C)[50]. An MITF peak is present within the gene body of *PTEN* (intron 4), and a second one is 114 kb downstream of the *PTEN* transcriptional start site (TSS) (i.e., +114 kb). Both MITF peaks were centered on an M-box motif (5′-TCATGTG-3′). At both intron-4 and +114 kb MITF peaks, the H3K27Ac signal was unchanged in ΔMITF mutant cell lines. However, six distal enhancers (painted light blue) exhibited a twofold greater H3K27ac signal in MITF mutant cell lines compared to wild-type cell lines (+140 kb, +210 kb, +230 kb, +253 kb, +301 kb and +317 kb), suggesting increased transcription of *PTEN* in the former. We confirmed the +140 kb MITF peak after performing a quantitative ChIP experiments on 501Mel cells expressing MITF-HA (Fig. 6D, E), using Tyrosinase (TYR) and PRM1 as positive and negative controls respectively. Overall, reduction of MITF results in an increase in PTEN mRNA expression. As such, it is plausible that Brn2 induces *Pten* transcription through two different, but concurrent mechanisms: (i) directly through BRN2 binding to the *PTEN* promoter to induce its transcription and (ii) indirectly via BRN2 modulating (repressing/inducing depending

of the situation) MITF expression, with MITF binding to the 3′ end of *PTEN* to inhibit its transcription. In the absence of BRN2, these mechanisms are disrupted and *PTEN* transcription is downregulated in our conditions. In conclusion, MITF represses directly *PTEN* transcription.

Overall, our results are consistent with a model in which reduction of BRN2 reduces PTEN transcription in vitro and in vivo, thus ramping up PI3K signaling and inducing both the initiation of melanoma and the formation of metastases.

## Discussion

A well-established principle of cancer biology is that tumors are initiated by a combination of oncogene activation together with loss of tumor suppressor expression or activity. In melanoma, key oncogenic drivers, such as BRAF and NRAS, have been well defined. Loss of P16 or PTEN tumor suppressor activity is required to bypass oncogene-induced senescence and permit melanoma initiation. However, while inactivation of tumor suppressors by mutation has been extensively studied, it is less well understood how their activity may be modulated by changes in their mRNA expression mediated by key melanoma-associated transcription factors. At this point, it is important to note that factors involved in melanoma initiation will not automatically reflect/translate into overall survival. Here, we identify BRN2, a key transcription factor lying downstream of three melanoma-associated signaling pathways (WNT/β-catenin, MAPK, and PI3K), as a tumor suppressor that functions to regulate PTEN expression. Thus, monoallelic loss of *Brn2* promotes melanoma initiation in a *Braf*$^{V600E}$/*Pten*$^{F/+}$ background where mono-allelic loss of *Pten* sensitizes cells to loss of *Brn2*.

Previous work has primarily linked BRN2 to melanoma migration and invasion in vitro and in xenograft experiments[13,20,42,51], but its role during melanoma initiation and proliferation in vivo and in normal melanocytes had not been determined. We report that, consistent with *BRN2* playing a key role as a tumor suppressor in melanomagenesis, its locus is frequently lost in human skin cutaneous melanoma (SKCM) metastases, independently of their *NRAS* or *BRAF* status, and that *BRN2* status contributes to overall patient survival. Significantly, the overall survival of patients with a mono-allelic loss of PTEN is higher when the *BRN2* locus is intact.

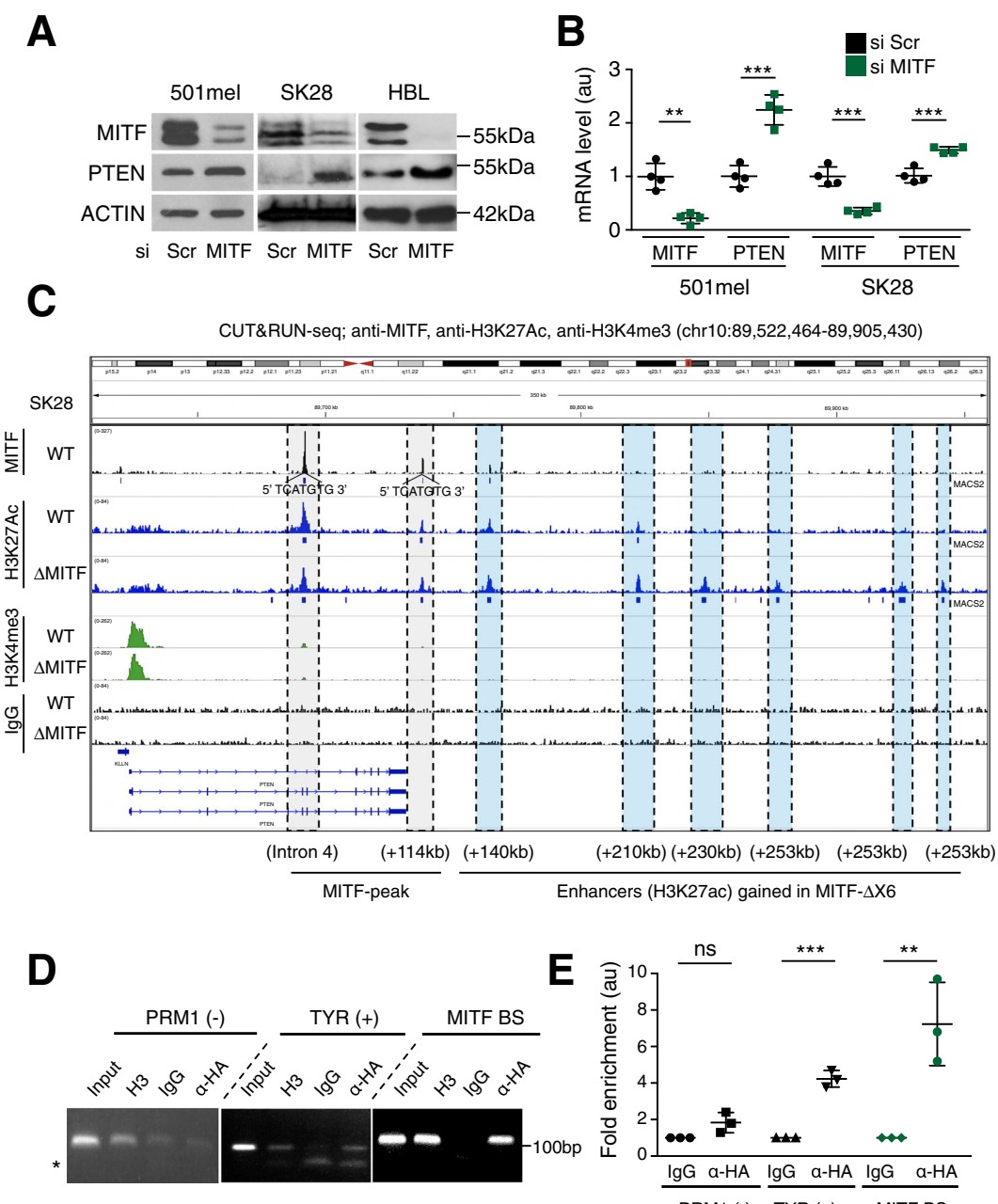

Although these observations are consistent with BRN2 affecting human melanoma initiation and progression, loss of the *BRN2* locus is frequently associated with large segmental deletions that affect the long arm of chromosome 6 (6q) as it was observed in 14 out of 53, and 9 out of 32 out of primary melanoma and in 17 out of 21 melanoma cell lines[52–54], and confirmed in this study on a total of 205 out of 338 melanomas, and 11 out of 25 melanoma cell lines. It has already been shown that the loss of 6q was associated with a worse prognosis[52]. According to our observations, the loss of 6q is more detrimental for the overall survival than the focal loss of the *BRN2* locus (Fig. 1 and Supplementary Fig. 1). Moreover, in vitro studies have shown that several genes are linked to melanomagenesis in the co-deleted region, including *ARID1B*, *CCNC*, and *ROS1*, although none have yet been shown to be functionally important in melanoma in vivo[48,55–61]. Similarly, we analysed the SKCM TCGA study to evaluate the overall survival comparing the mono allelic loss and the diploid state for *CCNC*, *ROS1* and *ARID1B* (Fig. S1).

There is no statistical significance for *CCNC* ($p = 0.086$), *ROS1* ($p = 0.27$), or *ARID1B* ($p = 0.66$).

Thus, it might be argued that any of the genes located in this frequently deleted region may be acting to modulate melanoma initiation or progression. However, our functional mouse molecular genetics models show conclusively that the heterozygous or homozygous loss of Brn2 promotes melanoma initiation and initial growth. Similarly, our transcriptomic analysis showed that the mRNA levels of the seven genes in the region co-deleted in humans (*Arid1b, Mchr2, Ccnc, Cdk19, Dll1, Ros1,* and *Crybg1/Aim1*) were not affected in the mouse tumors (Braf-Pten-Brn2-WT, Braf-Pten-Brn2-het and Braf-Pten-Brn2-hom). Collectively, these results strongly suggest that the reduction/loss of Brn2 levels is a critical event that cooperates with heterozygous Pten to promote the initiation and growth of melanoma, independently of the co-deleted genes in melanoma. Finally, independent initiation events are promoted when the level of Brn2 is lower than normal -heterozygous or homozygous- since the number of

**Fig. 6 MITF binds downstream of *PTEN* gene, MITF loss induces its transcription, and enhancers flanking *PTEN* are activated in MITF-depleted cells. A** Western blot analysis of MITF and PTEN from human melanoma cells (501mel, SK28, and HBL) after siRNA-mediated knockdown. Actin was used as a loading control. A representative western blot is shown, raw data are presented in Supplementary Fig. 8. The molecular weight is indicated in kDa. Scr = Scrambled. **B** RT-qPCR of MITF and PTEN from human melanoma cells (501mel and SK28) after siRNA-MITF and Scr knockdown. All values were normalized against TBP. The analysis was performed on three independent experiments with technical triplicates, au = arbitrary unit. **C** Screenshot of IGV genome browser (GRCH37/hg19) visualization of MITF, H3K27Ac and H3K4me3 binding to the *PTEN* locus in SkMel28 cell lines that are MITF-WT or mutant (i.e., MITF-ΔX6 = ΔMITF). Blue boxes below MITF and H3K27Ac tracks: signal above IgG background (i.e., peaks) called by MACS2. PTEN and downstream regions are shown, blue arrows indicate strand orientation and horizontal rectangles the exons. Y-axes are scaled per antibody sample. Anti-MITF CUT&RUN peaks present in WT cells that harbor an M-Box binding motif are painted gray. Six distal enhancers (painted light blue) exhibited a twofold greater H3K27ac signal in MITF mutant cell lines compared to wild-type cell lines. At least two CUT&RUN biological replicates were performed for MITF, H3K27ac, and H3K4me3. **D** ChIP assays of MITF binding downstream of *PTEN* in 501mel human melanoma cells stably expressing HA-Tagged MITF (location + 140 kb). ChIP assays are performed using an antibody against HA and analyzed after a 30-cycle PCR (exponential phase). The tyrosinase promoter (*TYR*) and *PRM1* were used as positive and negative controls, respectively. Input represents approximately 3% of the input used for the ChIP assay. H3 (histone H3) and IgG (Immunoglobulin G) were used as positive and negative technical controls for each region of interest, respectively. The oligonucleotides, their positions on the genome, and sizes of the amplified fragments are shown in Supplementary Tables 3 and 4. All data shown are representative of three independent assays. * corresponds to the remaining oligonucleotides. **E** Quantification of the ChIP–qPCR presented in (**D**) is plotted and normalized against IgG as a reference. au = arbitrary unit. PRM1 (−): PRM1 (negative control), TYR (+): tyrosinase promoter (positive control). Statistical analysis was performed using the two-tailed unpaired *t* test. Data are presented as mean values ± SD. ns = non-significant, *p < 0.05, **p < 0.01, and ***p < 0.001.

independent melanoma is higher in Braf-Pten-Brn2-het/hom mice than in Braf-Pten-Brn2-WT mice (Fig. 2B). Our observations are therefore consistent with BRN2 acting as a tumor suppressor in melanoma, and are in full agreement with the predominantly mutually exclusive pattern of BRN2 and Ki-67 in situ staining of invasive melanoma[20].

The presence of Braf[V600E] promotes proliferation prior to inducing senescence and the loss of Pten results in senescence bypass[3,34,37]. As such, we believe that the increased proliferation and tumor-initiation frequency observed in our *Braf[V600E]/Pten[F/+]* model arising as a consequence of the reduction of Brn2, is likely to occur as a consequence of the ability of Brn2 to activate Pten expression and suppress PI3K signaling either directly or potentially indirectly via Mitf repression during early melanomagenesis. In other words, inactivation of Brn2 or a reduction in its expression would lead to low expression of the remaining Pten allele and as a consequence increase the probability of senescence bypass. This explanation fits with the fact that in the absence of Pten on a Braf[V600E] background, the appearance of the tumors is too rapid to observe any difference between Brn2 + /+, Brn2 F/ + or Brn2 F/F. According to our model the transcription regulation by Brn2 and Mitf would not affect the level of Pten since it is already lost. Moreover, in a context in which *Pten* is diploid (WT) and Brn2 is reduced (het or hom), the downregulation of Pten would be limited and not sufficient to act efficiently as a tumor-suppressor.

Although it might be argued that the increase in visible tumor number in a Braf-Pten-Brn2-het and -hom might be a consequence of the increased proliferation caused by reduction/loss of Brn2, as evidenced by the increased proportion of Ki-67 positive cells within tumors, we feel this is unlikely. For welfare reasons the Braf-Pten-Brn2-WT mice were euthanized with tumors (total volume 2 cm³) after 4 weeks with about 8 tumors per mouse; by contrast the Braf-Pten-Brn2-het/hom mice were euthanized after 1.3 weeks with about 17 tumors per mouse of a similar size to the WT. Since tumor size is similar in the WT and Brn2 mutants, this indicates that the total number of cells in the WT and mutant tumors is similar and have undergone a similar number of cell divisions (though this occurred in a shorter time in the mutant). Since the WT and mutants have undergone a similar number of cell divisions, if proliferation were responsible for any increase in number of visible tumors then at 4 weeks the numbers of tumors in the WT should be the same as in the mutant at 1.3 weeks, whereas in fact, the tumor numbers in the WT were

around 50% of those in the Brn2 mutants. Although we do not want to rule out a contribution of proliferation, the more likely explanation is that the reduction of BRN2 promotes the bypass of senescence (by reducing the level of Pten) or/and promotes survival of proliferating melanoma cells at the early stages of initiation.

Our in vivo data therefore reveal that melanocyte-specific Brn2 reduction in Braf-Pten mice promotes the initiation and progression of melanoma. Melanoma initiation is promoted after proliferation is induced through various proteins including Mitf and senescence bypassed in this case through reduction of the level of Pten. Melanoma progression is induced by promoting invasion when the level of Brn2 is intermediate after inducing Axl and modulating the immune system. In the future, we will have to evaluate the kinase activity of Axl in this context, and the consequences of its inhibition *in cellulo* and in vivo to understand the Mitf/Brn2/Axl *ménage à trois*. Altogether, we establish Brn2 as a central tumor suppressor, acting at different steps of melanomagenesis, and complement numerous other studies showing the effect of Brn2 on invasion[13–15,20,29,51].

*Brn2* heterozygous mice were more prone to form LN metastases efficiently than mice that were *Brn2* WT when Pten was already heterozygous. The efficiency of the metastasis process depends on the status of *Brn2* as the number and size of each micro-metastasis was greater in Brn2 heterozygous than Brn2 wild-type melanoma. This is important as mono-allelic loss of BRN2 in human, corresponding to *BRN2* heterozygous melanoma, occurs in 53% of human melanoma. In *Brn2* wild-type melanoma, the change of the level/activity of Brn2 is possible but the amount of protein found in Brn2 wild-type melanoma absorbs better transient Brn2 depletion than in *Brn2* heterozygous melanoma.

Some residual Brn2 activity might be required for efficient melanoma progression. On one hand, Braf-Pten-Brn2-het and Braf-Pten-Brn2-hom melanomas proliferate faster, and on the other hand, according to our results (Fig. 4F,G) Braf-Pten-Brn2-hom melanomas have a reduced ability to invade compared to Braf-Pten-Brn2-het melanomas. Moreover, the lack of BRN2 reduces migration and increases the rate of apoptosis and/or anoikis[20,28,43]. In a more speculative way, one may think that optimal melanoma progression may be associated with a series of proliferative and invasive phases. In the absence of BRN2, cells are "fixed" in one stage and cannot switch from "proliferative" to "invasive" states. These cells are mainly proliferative and poorly

invasive as we observed in Braf-Pten-Brn2-hom situation. In the presence of Brn2 as WT [two alleles] or heterozygous [one allele], the corresponding mRNA and protein concentrations can be positively and negatively modulated by external/internal factors. The modulation of the level of Brn2 is more sensitive on a heterozygous background than on a WT background.

Altogether, one may understand that melanoma grows faster and forms melanoma when Brn2 expression is not too high (proliferation handicap) but at the same time not too low (migration/invasion/survival handicap). Indeed, on the one hand, Brn2-het and Brn2-hom melanomas proliferate faster, and on the other hand, Brn2-hom melanomas are handicapped to invade (Fig. 4F,G), to migrate[20] and to die by anoikis[28].

The formation of metastases is a multistep process in which cells proliferating in the primary tumor, and surviving the metastatic process, must undergo a switch to an invasive phenotype prior to a switch to a proliferative phenotype on site. Since the switch from proliferation to invasion, and back, has been associated with the activity of MITF, it is possible that for efficient metastatic colonization cells must be able to modulate MITF expression via BRN2. In this respect, it is especially important to note that BRN2 has been identified as a key regulator of MITF[13,49]. Consistent with the observation that MITF and BRN2 are frequently observed in mutually exclusive populations in melanoma, and that BRN2 may act in vivo as an MITF repressor[13], we have observed that in a non-tumoral context, the specific knock-out of Brn2 in vivo in melanocytes increases the level of Mitf (publication in preparation). Importantly, during progression of mouse Braf-Pten melanoma Mitf levels are modulated. During the initial phase of growth, melanoma cells are pigmented, indicative of Mitf activity, but later in situ they lose pigmentation and ability to produce Mitf[62]. Although the reduction of Brn2 in Braf-Pten-Brn2 primary melanoma is not sufficient to re-induce Mitf mRNA and pigmentation in all cells of these primary melanomas, the Braf-Pten melanoma cells that formed LN metastases were pigmented and re-expressed Sox10, a key transcription activator of Mitf. Thus, during progression of Braf-Pten melanomas Mitf is produced during initial growth and subsequently repressed during the second phase, before being re-expressed in LN metastases. It seems therefore likely that one role of the residual BRN2 in the heterozygotes may be to facilitate the modulation of MITF expression during metastatic spread.

It has been shown that BRAF and PI3K induce BRN2[15,16]. In consequence, it was expected that the level of BRN2 would decrease in the presence of such inhibitors. In addition, in the presence or absence of BRN2, melanoma cells are either more resistant or more sensitive to BRAF inhibitors, respectively[28,43]. This sensitivity would be associated with the function of BRN2 in DNA repair[43].

Although here we have focused on the role of BRN2 in melanoma, BRN2 is also expressed in a number of other cancer types including small cell lung cancer, neuroblastoma, glioblastoma and neuroendocrine prostate cancer[8–10]. While BRN2-mediated regulation of MITF is not likely to be important for non-melanoma cancers that do not express MITF, the ability of BRN2 to modulate PTEN expression uncovered here may play an equally important role in promoting the initiation and progression of these cancer types. In this respect the inducible knockout mice described here may represent an important tool to examine the role of BRN2 in non-melanoma cancers.

In conclusion, our results identify Brn2 as a key tumor suppressor through its ability to modulate Pten expression that, given the high prevalence of monoallelic mutations, is likely to play a key role in initiation of human melanoma and likely other BRN2-expressing cancer types. Since BRN2 expression is activated by PI3K signaling via PAX3[16], its ability to suppress PI3K signaling

by increasing PTEN expression may also provide cells with a negative feedback loop to control the PI3K pathway. Moreover activation of BRN2 by MAPK signaling downstream BRAF[15] as well as WNT/β-catenin signaling[14], may also permit coordination between these pathways and the PTEN/PI3K axis. Finally, given the importance of BRN2 in melanomagenesis identified here as well as its frequent heterozygosity, it may be important to further explore whether tumors with low BRN2 expression may be more susceptible to PI3K pathway inhibition, as it was shown for MAPK inhibitors. In this respect, this mouse model of BRN2-deficient melanoma could be useful for the pre-clinical testing of inhibitors for clinical development especially since it has been shown that BRN2 is involved in DNA repair[43].

## Methods

**TCGA data mining**. All TCGA data sets for somatic mutations, copy number alterations (CNAs), RNA levels, and clinical data for skin cutaneous melanoma and other cancers were retrieved from http://www.cbioportal.org on August, 2019. CNAs were calculated using the GISTIC algorithm. Samples with GISTIC copy-number values of "−1" were considered to have mono-allelic loss and those with GISTIC copy number values of "−2" to have bi-allelic loss. GISTIC copy number values > "+1" were considered as amplification. The TCGA datasets used were the: CNA-data set (n = 367), Seq-Data set ($n = 473$), Clark level data set ($n = 461$), and Breslow index data set ($n = 316$). mRNA levels were calculated from RNA sequencing read counts using RNA-Seq V2 RSEM and normalized to transcripts per million reads (TPM).

**Copy number analysis**. Copy number data for the regional metastatic cohort used in Fig. 1D was obtained from a previous study[63]. Briefly, DNA sequencing data of 1,500 cancer genes were used to generate copy number data. Copy number log ratios of sequenced exons were generated from bam files of tumor–normal pairs using CONTRA 2.03[64], with default parameters. Exons with insufficient coverage in the normal sample were removed. Copy number data were segmented using GLAD[65]. We used cut-off of −0.3 to determine mono-allelic loss of BRN2.

Copy number data with matched nevus-melanoma pairs used in Supplementary Fig. 1D was obtained from a previous study[32]. Log2 copy ratio values were processed using the PureCN R package[66] to estimate tumor purity and copy number.

**Level of expression of BRN2 mRNA in melanoma patients**. BRN2 mRNA levels were obtained from RNA sequencing data analyzed using the RNA Seq V2 RSEM pipeline as transcripts per million reads (TPM).

**Mouse models**. Mice were bred and maintained in the specific pathogen-free mouse colony of the Institut Curie, in accordance with the institute's regulations and French and European Union laws. Mice were bred and maintained in the specific pathogen-free mouse colony of the Institut Curie, in accordance with the institute's regulations and French and European Union laws. The transgenic *Tyr::CreERT2* (031281—B6N.Cg-Tg(Tyr-cre/ERT2)1Lru/J), *BrafV600E/+*, *Pten* (006440 - B6.129S4-Ptentm1Hwu/J) and *Brn2* mice have been described and characterized elsewhere[33–35,38,67]. All mouse lines were backcrossed onto a C57BL/6 background for more than ten generations. All desired combinations of genotypes were obtained through crosses. Mice were born with the expected ratio of Mendelian inheritance and no changes in gender ratios were observed. Experimental mice were of both genders and no apparent phenotypic differences between genders were observed. No statistical methods were used to predetermine sample size. The sample size was sufficient to measure the effect size for all experiments presented in this study. The experiments were not randomized, and the investigators were not blinded to allocation during the experiments and outcome assessment. Mice were housed in a certified animal facility with a 12-hour light/dark cycle in a temperature-controlled room (22 ± 1 °C) with free access to water and food.

**Growth of the mouse melanoma cell lines in syngeneic mice**. C57BL/6 mice were purchased from Charles River Laboratories. Twenty-four C57BL/6 mice were injected with Brn2 + /+ [m6 and m50], Brn2 F/ + [m36 and m59], and Brn2 F/F [m8 and m82] mouse melanoma cell lines. The cells were resuspended in PBS and 10⁵ cells (100 μL) subcutaneously implanted into the flanks of seven-week-old C57BL/6 mice using a 27-gauge needle. Presence of tumors was detected from day 15 to 50, it was independent of the genotypes.

**Ethical rules**. Animal care, use, and experimental procedures were conducted in accordance with recommendations of the European Community (86/609/EEC) and Union (2010/63/UE) and the French National Committee (87/848). Animal care and use were approved by the ethics committee of the Curie Institute in compliance with the institutional guidelines. Experimental procedures were specifically

approved by the ethics committee of the Institut Curie CEEA-IC #118 (CEEA-IC 2016-001) in compliance with the international guidelines.

**Mouse genotyping.** Mouse biopsies were digested overnight at 55°C using 200 ng proteinase K (Roche, #11 243 233 001) in 500 µL lysis buffer containing 16 mM [NH4]2 SO4, 67 mM Tris-HCl [pH 8.8 at 25 °C], 0.01% [v/v] Tween-20, in deionized H$_2$O. Proteinase K was inactivated for 20 min at 95 °C. Primers and PCR conditions are described (Supplementary Tables 3 and 4). PCR products were separated by agarose (Invitrogen, #15510027) gel electrophoresis. Genotyping were performed accordingly (Supplementary Tables 3 and 4).

**In vivo gene activation/deletion and melanoma monitoring.** Newborn mice were treated dorsally with 20 µL/day/mouse tamoxifen (Sigma, T5648, working concentration 20 µg/mL in DMSO) for the first three consecutive days after birth. Non-tamoxifen-induced mice of the same genotype were used as controls. After the application of tamoxifen, the mice were evaluated for the appearance of tumors and their progression once per week or more frequently if required. Developing skin excrescences > 3 mm diameter were considered to be melanomas, and validated after growth. Mice were euthanized and autopsied four weeks after tumor appearance or once the tumors reached 2 cm$^3$. Melanoma-specific survival curves were estimated from the day of euthanasia. Mouse melanomas were excised, rinsed in cold phosphate-buffered saline (PBS, Euromedex, ET330-A) and divided into four parts, two snap-frozen in liquid nitrogen for subsequent transcriptomic and western blot analysis and two fixed in 4% paraformaldehyde (PFA, Euromedex, 15714-S) and embedded in paraffin or OCT (VWR, #00411243) for histological analysis and immunostaining. Inguinal lymph nodes were fixed in 4% PFA and embedded in paraffin or OCT for histological analysis and immunostaining.

**Detection of defloxing from mouse melanoma tissue.** DNA extraction from paraffin-embedded melanoma sections (10 µm) was performed using the QIAamp DNA FFPE Tissue Kit (Qiagen, #56404), according to the manufacturer's instructions. A PCR using DNA extracted from each tumor was performed to verify the mouse genotype and proper defloxing of the modified genes (see Key Resource Table and Supplementary Tables 3 and 4 for PCR conditions and primers sequences).

**Immunohistochemistry of mouse melanoma and inguinal lymph nodes.** Paraffin-embedded mouse melanomas were sectioned into 7-µm-thick transverse sections and stained with hematoxylin/eosin (H&E), as previously described[68]. For immunostaining, sections were deparaffinized, rinsed in Tris-buffered saline (TBS; 20 mM Tris (Sigma: T-1503) pH7.6), 150 mM NaCl (OSI, A4321152), containing 0.1% [v/v] Tween-20 (VWR 8221840500) (TBST), depigmented with H$_2$O$_2$ (Sigma, H1009) for 15 min, boiled for 20 min in 10 mM sodium citrate (VWR, 1120051000), and blocked with TBST containing 3% bovine serum albumin (BSA, Sigma A9418). Sections were incubated overnight at 4 °C in TBST containing 3% BSA with antibodies against Ki-67 (Nova-Costra, NCL-Ki67p), BrdU (BD Biosciences, #555627), or PTEN (Cell signaling, #9559). The sections were then incubated in secondary biotinylated anti-rabbit or anti-mouse antibodies for 2 h at room temperature (RT). AEC (Sigma-Aldrich, A6926) was used to reveal bound antibody according to the manufacturer's instructions. All sections were counterstained with hematoxylin. Images were captured using a ZEISS Axio Imager 2 with Axiocam 506 color cameras. Image analysis was performed using ZEISS ZEN, Adobe Photoshop, and ImageJ software. Quantifications of Ki-67 and BrdU stainings were determined as a percentage. The percentage of Ki-67$^+$ and BrdU$^+$ cells from three fields (1000–2000 cells/field) from three independent tumors per genotype was determined and normalized.

**Immunofluorescence of mouse inguinal lymph nodes.** OCT-embedded lymph nodes were sectioned into 7-µm-thick transverse sections and rinsed in TBST, depigmented with H$_2$O$_2$ (Sigma, H1009) for 15 min, boiled for 20 min in 10 mM sodium citrate (VWR, 1120051000) and blocked with TBST containing 3% BSA (Sigma A9418). Sections were incubated overnight at 4 °C in TBST containing 3% BSA with antibodies against SOX10 (Abcam, ab155279). Sections were then incubated in secondary donkey anti-mouse 647 (Abcam, ab150107) and goat anti-rabbit IgG (H&L)-Affinity Pure, DyLight® 650 Conjugate (GtxRb-003-E650NHSX) antibodies for 2 h at RT. All sections were counterstained with DAPI (0.5 µg/mL in ethanol, Sigma, D9542). Images were captured using a ZEISS Axio Imager 2 with Axiocam 506 color cameras. Image analysis was performed using ZEISS ZEN, Adobe Photoshop, and ImageJ software.

**Protein extraction from tumors.** All steps were performed at 4 °C. Tissues were transferred to a tube containing 2.8-mm stainless steel beads and 1 mL RIPA buffer supplemented with sodium orthovanadate (Sigma, S6508), Complete protease inhibitor (Roche, #11873580001) and Phostop (Roche, #04906837001) were added. Tissues were homogenized three times for 3 min using the BeadBug™ homogenizer at 4 °C, followed by a brief centrifugation at 17,949 g for 1 min at 4 °C. Supernatants were transferred and centrifuged for 20 min at 15,294 × g. Supernatants

were collected and incubated with 200 µL previously PBS-washed G-Sepharose beads (GE Healthcare, #17-0618-01) for 2 h. Samples were centrifuged at 15,294 × g for 5 min and quantified using the Bradford assay.

**Microarray analysis.** Only one tumor per mouse was considered and corresponding to the biggest one. Tumors had the same size since we harvested tumors for transcriptomic analyses when they reached a size of 1cm$^3$. RNA from mouse melanomas (5 Braf-Pten-Brn2-WT, 10 Braf-Pten-Brn2-het, and 10 Braf-Pten-Brn2-hom) and mouse melanoma cell lines (4 Braf-Pten-Brn2-WT, 2 Braf-Pten-Brn2-het, and 3 Braf-Pten-Brn2-hom) established from the mouse melanoma tumors was extracted using the miRNeasy Kit (Qiagen, #217004). RNA Integrity was assessed using an Agilent BioAnalyser 2100 (Agilent Technologies), only RNA with a RIN > 7 were kept for the analysis. RNA concentrations were measured using a NanoDrop (NanoDrop Technologies). Complementary RNA (cRNA) was synthesized using the GeneChip 3'IVT Plus reagent Kit (Thermofisher, #902415), according to the manufacturer's protocol. In brief, total RNA was first reverse transcribed using a T7-Oligo(dT) promoter primer for first-strand cDNA synthesis. After RNase H treatment and second-strand cDNA synthesis, the double-stranded cDNA was purified and served as template for subsequent in vitro transcription (IVT). The IVT reaction was carried out in the presence of T7 RNA polymerase and a biotinylated nucleotide analog/ribonucleotide mix for cRNA amplification and biotin labeling. The biotinylated cRNA targets were then cleaned up, fragmented, and 11 µg cRNA hybridized to mouse MOE430 gene expression Affymetrix microarrays (Affymetrix, #900443). After washing and staining, using the Affymetrix fluidics station 450 (Affymetrix, # 00-0079), the probe arrays were scanned using an Affymetrix GeneChip Scanner 3000 (Affymetrix, # 00-0210).

The microarray data were normalized using the RMA (Robust Multichip Average) function of the edgeR package[69]. For genes targeted by multiple probesets, only the most variant probeset was kept. Differentially expressed gene analysis was performed with R, for the cell lines and the tumors using the limma package[70] available from Bioconductor (http://www.bioconductor.org). An enriched gene-ontology (Gene-Ontology Biological-Process, 2018) and pathway (WikiPathways human, 2019 & KEGG human, 2019) analysis was performed on the 296 genes found overexpressed in the Braf-Pten-Brn2-het compared to the Braf-Pten-Brn2-WT using Enrichr[71]. Gene-set enrichment analysis (GSEA) was performed using the "Verfaillie" signature[44]. These signatures originally containing too much genes (>1,000) were reduced to fit to the GSEA algorithm by selecting the most differentially expressed genes based on their fold change, a threshold of 4 standard deviation was selected. For the tumors analysis the GSEA was ran with one thousand permutations gene set based. For the analysis of the cell lines, the run pre-ranked function of the GSEA software was used using one thousand permutations gene set based. The genes were ranked according to their statistic t coming from the differential analysis results. The enrichment score (ES) reflects the degree to which a given gene set is represented in a ranked list of genes. Calculation of the ES is based on walking down a ranked list of genes and adjusting a running-sum statistic based on the presence of absence of a gene in the gene set. The magnitude of the increment represents the correlation of the gene with the phenotype. P values were estimated by gene-based permutation. GSEA normalizes the enrichment score for each gene set to account for the variation in set sizes, yielding a normalized enrichment score (NES). Only gene set with a NES absolute > 1.7 and a FDR < 0.01 were considered.

**Cell lines.** Melan-a and C57BL/6 9 v cells were grown in Ham's F12 media (GIBCO 11765054) supplemented with 10% fetal calf serum (FCS, GIBCO, 10270106), 1% Penicillin-Streptomycin (GIBCO, 15140), and 200 nM TPA (Sigma, P 8139)[4]. 501mel, 501mel-MITF-HA, HBL, SK-Mel-28, SK-Mel-28 Δex6, and Dauv-1 cells were grown in RPMI 1640 media (GIBCO, 11875101) supplemented with 10% FCS (GIBCO, 10270106) and 1% Penicillin-Streptomycin (GIBCO, 15140)[50,72–74]. SK28Δex6 melanoma cells lacking Mitf was previously produced[50]. Cells are routinely tested for the absence of mycoplasmas using MycoAlert (Lonza). Mouse melanoma cell lines were established as previously described[75]. The genetic status and the level of expression of key genes of these cell lines is given in Supplementary Table 1.

**Genomic DNA extraction from cell culture.** Genomic DNA was extracted from melanoma cell lines using the AllPrep DNAMini Kit (Qiagen, #80204) or QIAamp Kit (QIAGEN) according to manufacturer's instructions. The DNA region for BRAF and NRAS was amplified by PCR and submitted for sequencing (see for primers and conditions in the Key Resource Table and Supplementary Tables 3 and 4).

**siRNA-mediated knock-down.** siRNA targeting human BRN2 and MITF was purchased from Dharmacon as a SMART-pool mix of four sequences. siRNA targeting PTEN was purchased from Santa Cruz Biotech. Si Scramble (siSCR), with no known human or mouse targets, was purchased from Eurofins Genomics (Supplementary Table 5). All sequences and product references are listed in Supplementary Table 6. Briefly, cells were transfected with Lipofectamin2000 with 200 pmol siRNA or siSCR and assayed for mRNA expression or protein content 48 or 72 h post-transfection.

**Cell counting**. Phase-contrast pictures of cells were taken using a Zeiss Axiovert 135 microscope with an Axiocam MRC camera. Cells were counted using a LUNA automated cell counter (L10001) and cell-counting slides (L12001).

**Clonogenic assays**. For clonogenic assays, six-well tissue culture plates were seeded with 500 cells in complete medium. The medium was changed 24 h after cell seeding and replaced with complete medium containing the indicated concentrations of Binimetinib (Selleck), MK-2206 2HCl (Selleck), LY294002 (Calbiochem) or PLX4720 (Axon Medchem). After 9 days of incubation, colonies were fixed with 4% PFA (paraformaldehyde), stained with Crystal violet in 10% ethanol, and counted on images. IC50 were determined for each pharmacological agent and for each cell line using the number of colonies in mock treated condition as the top response. Cell lines have been treated with Binimetinib from $10^{-3}$ to 100 μM, with PLX4720 from $10^{-3}$ to 100 μM, with LY294002 from $5.10^{-3}$ to 50 μM and with MK-2206 from $10^{-3}$ to 10 μM. We used the resulted sigmoidal curves to calculate the IC50 with Graphpad Prism. Experiments were performed at least in triplicate.

**Western blotting and detection**. Whole-cell lysate was prepared from human melanoma cell lines using RIPA buffer supplemented with sodium orthovanadate (Sigma S 6508), complete inhibitor (Roche, #11873580001), and Phostop (Roche, #04906837001). SDS-PAGE was carried out on homemade 10% polyacrylamide protein gels. Following the transfer of the proteins, the nitrocellulose membranes were blocked in TBST with 5% non-fat dry milk for 1.5 h at RT and then probed with antibodies against BRN2 (Cell signaling, #12137), MITF (Abcam, ab12039), β-actin (Sigma, A5441), cyclin D1 (Cell signaling, #2926), PTEN (Cell signaling, #9559), phospho-S6 (Cell signaling, #4857), S6 (Cell signaling, #2317), phospho-AKT (Cell signaling, #3787), AKT (Cell signaling, #4685), or vinculin (Sigma, V9131). Primary antibodies were applied in TBST/5% non-fat dry milk overnight at 4 °C and visualized using secondary antibodies (HRP-conjugated goat anti-rabbit IgG, Jackson, 111-035-003 and HRP-conjugated goat anti-mouse IgG, Jackson, 115-035-003) in TBST/5% non-fat dry milk for 1 h at RT. Blots were incubated in ECL (Pierce, #34075) and revealed in the dark using ECL hyperfilm (GE Healthcare, RPN3103K). All primary antibodies were used at a dilution of 1/1,000, except β-actin and vinculin (1/5,000). All secondary antibodies were used at a dilution of 1/20,000. Pageruler (ThermoFisher, #26616) was used as the molecular marker. Quantification of the western blots was performed using *ImageJ* software. Quantification of the western blots was performed using *ImageJ* software. See Supplementary table 7.

**Chromatin immunoprecipitation**. ChIP experiments were performed as previously described[4], and see Supplementary Table 6. ChIP assays of BRN2 binding to the *PTEN* promoter. ChIP assays were performed using an antibody against BRN2 and analyzed after 30-cycle PCR in exponentially growing phase of Dauv-1 melanoma cells. *PAX3* promoter (prom.) and Brn2 coding sequences (CDS) were used as positive and negative controls, respectively. Input represents approximately 0.4% of the input used for the ChIP assay. H3 (histone H3) and IgG (Immunoglobulin G) were used as positive and negative controls for each region of interest, respectively. The oligonucleotides, their position on the genome, and the sizes of the amplified fragments are given in Supplementary Tables 3 and 4.

**RNA extraction and (ChIP) RT-qPCR**. Tissues were crushed with a mortar and pestle, and stainless steel beads. Qiazol was used to homogenize the samples prior extracting RNA using the miRNeasy Kit. Purified RNA was reversed transcribed using M-MLV Reverse Transcriptase. Real-time quantitative PCR (qPCR) was performed using iTaq™ Universal SYBR Green Supermix. Each sample was run in technical triplicates and the quantified RNA normalized against TBP (human) or Gapdh (mouse) as housekeeping transcripts (Supplementary Tables 3 and 4).

**CUT&RUN**. Anti-MITF, anti-H3K27Ac and anti-H3K4Me3 Cleavage Under Targets and Release Using Nuclease (CUT&RUN) sequencing was performed in SK28 melanoma cell lines that are MITF-WT or null (ΔMITF) as described[76], with minor modifications. Cells (~75–80% confluent) were harvested by cell scraping (Corning), centrifuged at $600 \times g$ (Eppendorf, centrifuge 5424) and washed in calcium-free wash-buffer (20 mM HEPES, pH7.5, 150 mM NaCl, 0.5 mM spermidine and protease inhibitor cocktail, cOmplete Mini, EDTA-free Roche). Pre-activated Concanavalin A-coated magnetic beads (Bangs Laboratories Inc) were added to 100uL cell suspensions ($2 \times 10^5$ cells) and incubated at 4 °C for 15 min. Antibody dilution buffer (wash-buffer with 2 mM EDTA and 0.03% digitonin) containing anti-MITF (Sigma, HPA003259, 1:100), anti-H3K27Ac (Millipore, 07-360, 1:100), anti-H3K4Me3 (Millipore, 05-745 R, 1:100) and Rabbit IgG (Millipore, 12-370, 1:100) was added and cells were incubated at 4 °C overnight. The next day, cells were placed on a magnetic rack and washed twice in dig-wash buffer (wash buffer containing 0.025% digitonin). pAG-MNase at a concentration of 500 μg/ mL was added and cells were incubated at 4 °C for 30 mins (pAG-MNase was purified in Dr. Robert Cornell's research group at the University of Iowa). The pAG-MNase reactions were quenched with 2X Stop buffer (340 mM NaCl, 867 20 mM EDTA, 4 mM EGTA, 0.05% Digitonin, 100 μg/ mL RNAse A and 50 μg/ mL Glycogen). Released DNA fragments were Phosphatase K (1 μL/mL, Thermo Fisher Scientific)

treated at 50 °C for 1 hr and purified by phenol/chloroform-extracted and ethanol-precipitated. Fragment sizes were analyzed using a 2100 Bioanalyzer (Agilent). All CUT&RUN experiments were performed in duplicate.

**Library preparation and data analysis**. CUT&RUN-seq libraries were prepared using the KAPA Hyper Prep Kit (Roche). Quality control post-library amplification was conducted using the 2100 Bioanalyzer (Agilent). Fragment analysis and fragments sizes were compared pre- and post-library amplification to insure correct size selection. Libraries were pooled to equimolar concentrations and sequenced with paired-end 100 bp reads on an Illumina HiSeq X platform. Paired-end FastQ files were processed through FastQC (Babraham Bioinformatics) for quality control. Reads were trimmed using Trim Galore Version 0.6.3 (Developed by Felix Krueger at the Babraham Institute). Bowtie2 version 2.1.0[77] was used to map the reads against the hg19 genome assembly and MACS2 Version 2.1.1.20160309.6 was used to call peaks. The mapping parameters and peak calling analysis was performed as previously described[76,78] using IgG samples as background control. The Deeptools function "BAMcoverage"[79] was used to generate normalized (–RPKM) BigWigs files for visualization on Integrative Genomics Viewer (IGV)[80].

**Luciferase assays**. Human ($-2,375$ to $+840$) and murine ($-2136$ to $+936$) Pten-promoter fragments were cloned upstream 56 bp of CMV promoter conferring a very weak basal activity upstream luciferase (miniCMV::Luc, VectorBuilder) to generate hsPTEN::Luc (#1282) and mmPten::Luc (#1283) reporter vectors. Cells at 70% confluence were transiently transfected in twelve-well plates, using 2μL of Lipofectamin2000 (Invitrogen), 500 ng of total plasmid DNA 200 ng Pten::Luc reporter plasmids (#1282 or #1283) or miniCMV::Luc reporter (#1281), 200 ng of the expression vectors CMV::EGFP-Brn2 (#896) or CMV::EGFP (#1042) (equimolar) as a control[27]; 100 ng of the HSV-TK::renilla luciferase construct (#894) in Opti-MEM medium (Gibco). Luciferase activity and renilla luciferase activity were determined 48 h after transfection. Luciferase activity was normalized against renilla luciferase activity.

**Software**. GraphPad PRISM, R version 3.6.3 (R Foundation for Statistical Computing, Vienna, Austria), Adobe Illustrator, Adobe Photoshop, and Microsoft Power Point software were used to analyze data and generate all graphs and figures.

**Quantification and statistical analysis**. Cell culture-based experiments were performed in at least biological triplicate and validated three times as technical triplicates. P values for the comparison of two groups were calculated using the unpaired Student t-test or Mann-Whitney test. P values for the comparison of multiple groups were calculated using the analysis of variance (ANOVA) and Fisher's least significant difference tests. P-values for categorical data were calculated using the Chi-square test. P-values for the comparison of Kaplan–Meier curves were calculated using the log-rank (Mantel–Cox) or Gehan-Breslow-Wilcoxon test giving weight to the early events. P values were reported as computed by Prism 6.

**Reporting summary**. Further information on research design is available in the Nature Research Reporting Summary linked to this article.

## Data availability

Microarray gene expression data that support the findings of this study have been deposited in Gene Expression Omnibus (GEO) with the accession codes GSE126524 and GSE163085. The TCGA Skin Cutaneous Melanoma data referenced during the study are available in a public repository from the National Cancer Institute (NCI) Genomic Data Commons (GDC) website (https://portal.gdc.cancer.gov). Cut&Run assay data that support the findings of this study have been deposited in Gene Expression Omnibus (GEO) with the accession codes GSE153020. The accession number for the sequencing data reported in this paper is dbGaP: phs001550.v1.p1. All the other data supporting the findings of this study are available within the article and its supplementary information/ data files and from the corresponding author upon reasonable request. A reporting summary for this article is available as a Supplementary Information file. The data that support the findings of this study are available from the corresponding author upon reasonable request. All details concerning antibodies, chemicals, critical commercial assays, cell lines, model organisms, oligonucleotides, and software and algorithms can be found in Supplementary Table 7a–g. Source data are provided with this paper.

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

## Acknowledgements

We are grateful to Dorothy C Bennett, Ghanem Ghanem, Florence Faure, and Meenhard Herlyn for providing cell lines. Hong Wu for providing Pten flox mice. We thank the Institut Curie staff responsible for the animal colony (especially P. Dubreuil), and the histology (S. Leboucher), FACS (C. Lasgi), and PICT-IBiSA imaging (C. Lovo) facilities. This work was supported by the Ligue nationale contre le cancer, INCa, ITMO Cancer, Fondation ARC (PGA), and is under the program «Investissements d'Avenir» launched by the French Government and implemented by ANR Labex CelTisPhyBio (ANR-11-LBX-0038 and ANR-10-IDEX-0001-02 PSL). M.H. had a fellowship from PSL and FRM, P.S. had a fellowship from INSERM, and M.L.C. had a fellowship from FRM. C.R.G. was supported by the Ludwig Institute for Cancer Research. RC was supported in part by a grant from the National Institutes of Health (AR062547, RAC) and a postdoctoral fellowship from the American Association for Anatomy (CK). E.S. was supported from the Research Fund of Iceland (184861-053). L.L. and E.S. are supported by a Jules Verne grant.

## Author contributions

M.H., P.S., V.P., J.R., V.D., M.L.C., C.K., M.P., F.R., N.C., A.S., L.S.C., L.M., M.L., and G.B.J. conducted the experiments M.H., P.S., R.A.C., C.R.G. and L.L. designed the experiments M.H., P.S., V.P., F.G., V.D., Z.A., A.S., A.B., I.D., C.R.G., and L.L. wrote and reviewed the manuscript C.R.G., and L.L. secured the funding D.M. and E.S. provided reagents I.D., A.S., C.R.G. provided expertise and feedback

## Competing interests

The authors declare no competing interests.
