## [Peer Review File · Nature Communications]

Reviewers' comments:

Reviewer #1 (Remarks to the Author):

In the manuscript "BRN2 is a non-canonical melanoma tumor-suppressor", the authors use a combination of TCGA data analysis, a novel mouse model, and in vitro cell line experiments to investigate how loss of BRN2 influences human and mouse melanoma progression. The main conclusions the authors draw are that haplo-insufficiency of BRN2 is sufficient for melanoma initiation, that haplo-insufficiency of BRN2 is sufficient for melanoma metastasis, that BRN2 loss increases PTEN expression, and that PTEN expression is increased through two novel molecular mechanisms. Although the manuscript does contain some well-conducted and interesting experiments, the majority of the authors' conclusions are unsubstantiated.

Strengths of the study

- 1) The authors have convincingly demonstrated an interesting, previously unknown, relationship between BRN2 loss and PTEN activation. Specifically, the authors demonstrate:
 - a. On a genetic background of activated BRAF and partial PTEN loss, partial BRN2 loss increases melanoma proliferation.
 - b. The increased proliferation correlates with increased AKT signaling and decreased PTEN.
 - c. BRN2 directly binds the PTEN promoter.
- 2) The mouse model the authors have generated is elegant and likely will be useful for the field.
- 3) Overall, this reviewer believes these observations are novel and would be of interest to the fields of melanoma genetics, melanocyte and neural crest development, and cancer cell signaling.

This reviewer encourages the authors to consider re-framing this study to focus on these strengths.

Weaknesses of the study

- 1) The analyses conducted with the TCGA data are flawed in logic, approach and statistical analyses. Over-all, this reviewer does not agree that the authors can conclude that "The BRN2 locus is frequently deleted in human melanoma and reduced BRN2 mRNA levels correlate with reduced overall survival and worsening of prognostic factors" as they claim.
 - a. Advanced melanomas have very high copy number variations (Bastian, et al (1998) Cancer Research, PMID: 9605762). Consequently, a large number of genes present with loss or disruption of a single allele by chance. BRN2 is located on Chr. 6q, which is among the most frequently lost chromosomal arms in melanoma (Guan, et al (1998) Cancer Genet Cytogenet, PMID: 9844599). Any gene on this arm will demonstrate the same pattern the authors present here. To make their argument, the authors must convincingly show that BRN2 is lost by focal deletions at a frequency significantly greater than all other genes on Chr. 6q.
 - b. The authors have selectively omitted data that refute their hypothesis. When this reviewer conducted the same analysis with the same TCGA data, the graphed frequencies were replicated (36% diploid, 53% Het, 1.1% deletion), however there were also 7.1% gain and 0.6% amplification. As the gain percentage is greater than the deletion percentage, the figure caption "BRN2 locus is frequently deleted in human melanoma" is false.
 - c. Similarly, the authors omit Clark's Level 1 in Figure 1E, and provide no rationale for the grouping of T1+T2 and T3+T4 in figure S2B. Further, the data distributions do not appear to be appropriate for application of the Mann-Whitney test. The authors should provide the raw data or statistical tests demonstrating that the shape of all distributions is the same. Overall, the data relating BRN2 mRNA expression to prognostic factors are unconvincing and should be substantiated by orthogonal studies.
 - d. The correlation between loss of 6q and survival has been very well established (Healy (1998) Oncogene PMID: 9619830). It is unclear whether the survival curves presented in Figure 1 provide novel information. A third category of "6q loss" should also be included, and compared to focal deletions of BRN2 and each other gene on this chromosomal arm.

2) Although the authors convincingly show with their mouse model that partial loss of BRN2 increases melanoma proliferation, this finding is expected from many previous publications, as the authors already discuss. The authors go on to claim that haplo-insufficiency is “sufficient” for both initiation and metastasis. However, these conclusions are not substantiated by the presented data:

- In the manner the experiments were conducted, the perceived increase in both initiation and metastasis could be explained by increased proliferation. For example, if the melanocytes proliferated faster (as the authors demonstrate) than at any given time point, the number of measurable tumors will be greater, even if the number of initiating events is the same. The authors should consider comparing the number of tumors per mouse at different time points for each background, when the average tumor size is the same.
- Similarly, the authors claim an observed increase in metastasis. However, the authors note that the number of metastatic events were actually the same. This is despite an increase in primary tumor number and size. The observed differences were the size of the lymph node and the number of melanocytes in the node – both of which could be explained by more rapidly dividing cells. A more reasonable conclusion is that the number of metastatic events is the same upon BRN2 loss (or even less relative to total primaries), but that the BRN2 het melanocytes proliferate faster. The authors should consider quantifying distal metastases, which the activated BRAF, PTEN deleted mice are known to induce. If partial loss of BRN2 can substitute for the loss of the second PTEN allele, the expectation is that activated BRAF, partial PTEN loss, partial BRN2 loss mice will also present with distal metastases.
- The authors only observe a difference on the specific background of activated BRAF and PTEN partial loss, but neither PTEN diploid nor PTEN full loss. This is a niche circumstance, the relevance of which to human melanoma is unclear, as PTEN loss generally occurs quite late during human melanoma progression. In either case, the conclusions of BRN2 loss being “sufficient” for any of these phenotypes are misleading. The authors also chose not to present the PTEN full loss data, nor comment on BRN2 full loss data, both of which are critical controls.

3) The data to support the claim of two novel mechanisms linking BRN2 loss to PTEN expression are preliminary.

- The microarray results provided are under-powered. In the methods the authors state that 13 tumors were profiled. In the presented results, only 7 total were used, inclusive of 5 experimental and 2 controls. What was the rationale for omitting the other 6 tumors? Were the samples representative of independent mice? Why were only 2 controls used? That the study was under-powered is clear from Table S3 which shows that no genes were significantly altered in expression when considering the adjusted p value.
- The relationships between either BRN2 or MITF and PTEN are both novel and of interest to the community. These studies could be strengthened by performing the experiments in more than one human cell line, by using a luciferase reporter to determine whether transcription factor binding increases or decreases transcription, and by performing both the BRN2 and MITF experiments in the same sets of cells. As is, it is unclear how the MITF data presented in figure S8 are connected with the rest of the manuscript.

4) The title, abstract, introduction and discussion contain misrepresentations of the previous literature and exaggerations of the results.

- Although the authors cite the majority of relevant work, this reviewer disagrees with the representations of the data within those studies and suggests a more careful reading and comprehensive assessment of the cited papers. Specifically, the contribution of BRN2 to melanoma metastasis has been looked at in vivo by multiple groups as has the consequence of BRN2 inhibition in normal melanocytes, and the relationship between MITF and BRN2 is more complex than suggested.
- Many of the conclusions drawn are exaggerated and need to be toned down. In particular, conclusions based upon the TCGA analyses and microarray data should be omitted unless the issues outlined above are addressed.

Reviewer #2 (Remarks to the Author):

In this study the authors argue that BRN2 is a haplo-insufficient tumor suppressor whose loss promotes melanoma initiation and progression. The mechanistic basis for this is proposed to be BRN2 regulation of PTEN transcription, either directly or via repression of MITF. Whereas previous studies have indicated that BRN2 promotes melanoma invasion (e.g. Goodall et al., 2008; Fane et al., 2017), the role of BRN2 in melanoma initiation and progression remains to be fully investigated. This report provides important clues and contributes to a better understanding of the role of this transcription factor in melanomagenesis and possibly other BRN2-dependent tumors

The BRN2 locus shows frequent copy number loss in cutaneous melanoma metastases, although these deletions also affect many other genes implicated in melanomagenesis (Fig. 1A-B). Additional data supporting that BRN2 is the key contributor to melanoma initiation and progression would significantly strengthen the manuscript.

Major issues

Mono-allelic loss of BRN2 is associated with worse overall survival. Considering that central thesis is that BRN2 is a tumor suppressor whose monoallelic loss promotes melanoma initiation and progression, frequency of BRN2 loss (in addition to mRNA levels) at early/pre-metastatic stages of melanoma should be included to further support the weak association BRN2 low vs. high mRNA levels with survival (Fig. 1G). Is this an early event in melanomagenesis?

Mouse model indicates that mono-allelic loss of BRN2 in neonatal mice promotes melanomagenesis. Recommend further experiments to support BRN2 tumor suppressive function. Does re-expression of BRN2 in cells that have lost BRN2 suppress (i) colony formation in vitro (ii) tumor formation in mice?

Would be useful to show survival data for the different cohorts of mice as well as histology of the primary tumors including staining with melanoma and proliferation markers.

Do tumors progress to metastatic melanoma beyond LN infiltration?

What effects does bi-allelic loss of BRN2 in melanoma cells have on tumorigenicity?
What is the impact of silencing the other genes in Chr6 (e.g. Arid1A, ROS)?

Evidence from clinical samples of melanoma in situ/invasive melanoma vs. adjacent precursor nevi suggests that BRN2 levels increase during transition to invasive melanoma (Zeng et al., 2018). Please discuss this and address whether mono-allelic loss of BRN2 occurs early in melanoma development.

All main figure in vitro analyses rely on comparisons of one transformed cell line and one non-transformed melanocyte (e.g. following BRN2 knockdown; Fig. 6D). For greater generalizability please include additional cell lines such as those in Fig. S8 and include cells with loss of PTEN.

Authors suggest that BRN2 may induce PTEN transcription partially by repressing MITF. While references for the link between BRN2 and MITF are cited, no evidence is presented to show that BRN2 inhibits MITF expression in these models or that BRN2 knockdown/loss enhances MITF expression. PTEN levels are instead monitored following direct knockdown of MITF. Include data showing regulation of MITF by BRN2 and corresponding PTEN levels.

To demonstrate that BRN2 loss can also repress PTEN indirectly via the induction of MITF, concomitant silencing of BRN2 and MITF should be performed. Likewise, MITF levels could be assessed in BRAF/PTEN/BRN2 tumors.

MITF promotes proliferation, whereas low MITF is associated with an invasive, slow-cycling phenotype. Could MITF also be contributing to the effects of Brn2 loss on proliferation and LN infiltration?

Since the overall conclusion of the study is that “BRN2 loss reduces PTEN transcription in vitro and in vivo, thus ramping up PI3K signaling and inducing both the initiation of melanoma and the formation of metastases” it would be interesting to determine the effects of BRAFi+PI3Ki inhibitors on BRAF/PTEN/BRN2 tumor initiation and LN metastasis.

Minor issues

BRN2 mRNA levels are lower in invasive melanomas (T3+T4) than localized. Please incorporate Fig. S2B into Fig. 1 as this shows the most clinical relevant staging (Clark’s level superseded by AJCC staging).

What is PTEN mutation status in cell lines used?

Based on the phenotype elicited by loss of Brn2, the authors state that “Brn2 acts as a tumor suppressor in vivo, and its loss induces melanoma initiation and increases tumor growth rate”; reword to more precisely indicate that loss of Brn2 in the context of monoallelic PTEN...

“Cell lines express Pten and Brn2 mRNA...(data not shown)” – include the data in supplemental figures.

“Cell number was monitored” after BRN2 knockdown (Fig. 4D). Cell cycle analysis to assess relative proportion of cells proliferating would further support the conclusions.

Transcriptomic analysis of tumors from BRAF-PTEN-BRN2 vs. BRAF-PTEN-WT (Fig. 5). Please indicate selection criteria for which tumors were analyzed. Was tumor size matched?

Reduction in PTEN positivity upon BRN2 loss (mouse tumor IHC – Fig. 6A) is not reflected in Western blot levels (Fig. 6B). PTEN protein levels look comparable between WT and BRN2 when normalized to actin intensity. Provide densitometry values normalized to actin in Fig. 6B. Also probing for MITF is suggested.

Reviewers' comments:

Reviewer #1 (Remarks to the Author):

In the manuscript "BRN2 is a non-canonical melanoma tumor-suppressor", the authors use a combination
of TCGA data analysis, a novel mouse model, and in vitro cell line experiments to investigate how loss of
BRN2 influences human and mouse melanoma progression. The main conclusions the authors draw are
that haploinsufficiency of BRN2 is sufficient for melanoma initiation, that haplo-insufficiency of BRN2 is
sufficient for melanoma metastasis, that BRN2 loss increases PTEN expression, and that PTEN expression
is increased through two novel molecular mechanisms. Although the manuscript does contain some well-
conducted and interesting experiments, the majority of the authors' conclusions are unsubstantiated.

Strengths of the study

1) The authors have convincingly demonstrated an interesting, previously unknown, relationship between
BRN2 loss and PTEN activation. Specifically, the authors demonstrate:

a. On a genetic background of activated BRAF and partial PTEN loss, partial BRN2 loss increases melanoma
proliferation.

b. The increased proliferation correlates with increased AKT signaling and decreased PTEN.

c. BRN2 directly binds the PTEN promoter.

2) The mouse model the authors have generated is elegant and likely will be useful for the field.

3) Overall, this reviewer believes these observations are novel and would be of interest to the fields of
melanoma genetics, melanocyte and neural crest development, and cancer cell signaling.

This reviewer encourages the authors to consider re-framing this study to focus on these strengths.

We thank the reviewer for these positive comments

Weaknesses of the study

1Q1

The analyses conducted with the TCGA data are flawed in logic, approach and statistical analyses. Over-
all, this reviewer does not agree that the authors can conclude that "The BRN2 locus is frequently deleted
in human melanoma and reduced BRN2 mRNA levels correlate with reduced overall survival and
worsening of prognostic factors" as they claim.

1A1.

We understand the reviewer's point of view. We built this study on the previous literature showing the
role of BRN2 in melanoma. The analysis presented in Figure 1 was performed as an exploratory correlative
analysis and was not meant to show definitive proof of Brn2's importance in melanomagenesis. Further
functional results presented in the rest of the manuscript including evidence obtained from the mouse
model of melanoma initiation and progression allowed us to generate solid conclusions on the role of
BRN2 *in vivo*. We modified the sentence to remove any ambiguity to "One BRN2 allele is frequently lost in
human melanoma and reduced BRN2 mRNA levels correlate with reduced overall survival."

1Q2

a. Advanced melanomas have very high copy number variations (Bastian, et al (1998) Cancer Research,
PMID: 9605762). Consequently, a large number of genes present with loss or disruption of a single allele
by chance.

BRN2 is located on Chr. 6q, which is among the most frequently lost chromosomal arms in melanoma
(Guan, et al (1998) Cancer Genet Cytogenet, PMID: 9844599). Any gene on this arm will demonstrate the
same pattern the authors present here. To make their argument, the authors must convincingly show that

BRN2 is lost by focal deletions at a frequency significantly greater than all other genes on Chr. 6q.

1A2.

One of the goals of this study, as the reviewer points out, was to generate a relevant mouse model for
human melanoma to evaluate the causal role of the loss of BRN2 during melanomagenesis, and especially
during initiation since it is the best way to address functionally this question. Indeed, and we fully agree
with the reviewer; Brn2 is located on chromosome 6q and this chromosomal arm is frequently lost.

As suggested by the referee we have added to our manuscript the ^{1,2} references since Bastian *et al*
analyzed 32 primary melanoma and Guan *et al* 21 melanoma cell lines. From the TCGA, our study
validates and solidifies the previous findings using a total of 367 melanomas.

As presented in Figure 1B some deletions are limited to the BRN2 locus, but indeed the vast majority
cover other loci including CCNC, ROS1 and ARID1B, which are more apical than the BRN2 locus. We re-
analyzed the SKCM TCGA study data using clinical annotations last updated August 8, 2019 to evaluate
overall survival comparing the mono allelic loss and the diploid state for BRN2, CCNC, ROS1 and ARID1B.
These data are shown in Figure 1C and Figure S1. It appears that at this stage of the follow up the clinical
study, statistical significance is not reached for BRN2 ($p = 0.067$), but for each of the other genes the
statistical significance is less significant than of BRN2 (CCNC, $p = 0.086$; ROS1, $p = 0.27$; ARID1B, $p = 0.66$)
However, using a different cohort (Figure 1D), statistical significance is reached for BRN2 ($p = 0.003$).
There is therefore a *trend* for BRN2, and potentially for CCNC, to demonstrate a lower overall survival
when mono-allelic compared with the diploid state. These trends were not observed for ROS1 and
ARID1B. In conclusion, after this more precise analysis the referees statement that 'Any gene on this arm
will demonstrate the same pattern the authors present here' is not correct.

Moreover, as the referee will appreciate, this study does not focus on CCNC, ROS1 and/or ARID1B, and
our goal was not to show that these genes were or were not of importance during melanomagenesis,
though in the future, it may be of importance to analyze *in vivo* the role of CCNC, ROS1 and ARID1B during
melanoma initiation.

The first goal of this analysis was not to show that BRN2 was specifically lost in human melanoma, as a
result of a focal deletion targeting BRN2, but to evaluate the number of BRN2 alleles that were lost in
human melanoma in a descriptive way. The second goal of this analysis was to evaluate whether this loss
was independent of BRAF or NRAS mutation status to better design our mouse melanoma model (see
Figure S2).

Importantly we demonstrate genetically in mice that the loss of Brn2 on a Braf/Pten background induces
more efficiently the number of melanomas per mouse. The mouse models revealed that the loss of Brn2
did not affect the level of expression of Ccnc, Ros1 and/or Arid1b in melanoma.

This study highlights the potential importance of Brn2 in melanomagenesis *in vivo* but it is not designed to
address the function(s) of the other genes on Chr.6q.

1Q3.

b. The authors have selectively omitted data that refute their hypothesis. When this reviewer conducted
the same analysis with the same TCGA data, the graphed frequencies were replicated (36% diploid, 53%
Het, 1.1% deletion), however there were also 7.1% gain and 0.6% amplification. As the gain percentage is
greater than the deletion percentage, the figure caption "BRN2 locus is frequently deleted in human
melanoma" is false.

1A3.

We understand the point of the referee. However, we did not omit this information, it was mentioned in
the original version (lines 13-14 p6) the other categories (gain and amplification). We have replaced the
caption of Figure 1A "The Brn2 locus is frequently deleted in human melanoma ..." by "One Brn2 allele is

frequently lost in human melanoma...". We also added in the figure legend "Gain and/or amplification of
the BRN2 locus (7.9%) are not displayed".

1Q4.

c. Similarly, the authors omit Clark's Level 1 in Figure 1E, and provide no rationale for the grouping of
T1+T2 and T3+T4 in figure S2B. Further, the data distributions do not appear to be appropriate for
application of the Mann-Whitney test. The authors should provide the raw data or statistical tests
demonstrating that the shape of all distributions is the same. Overall, the data relating BRN2 mRNA
expression to prognostic factors are unconvincing and should be substantiated by orthogonal studies.

1A4.

We did not want to omit Clark's Level 1 in the previous Figure 1E. Indeed Clark's levels are only relevant
for primary melanoma. In order to simplify our message, we decided to remove the panels associated
with the Clark's level and Breslow index (previous Figure S2B). It is important to note that modification of
melanomagenesis initiation will not automatically reflect/translates the OS. This point is added in the
discussion.

1Q5.

127 d. The correlation between loss of 6q and survival has been very well established (Healy (1998) Oncogene
PMID: 9619830). It is unclear whether the survival curves presented in Figure 1 provide novel information.
A third category of "6q loss" should also be included, and compared to focal deletions of BRN2 and each
other gene on this chromosomal arm.

1A5.

As the referee mentioned, loss of 6q and survival has been very well established. Therefore we did not
want to display these data. However, we evaluated the loss of 6q (n = 61) with the mono-allelic loss of
Brn2 (n = 132) (see Figure S1P). Indeed, it appears that the 6q loss is associated with a worse prognostic
than the BRN2 mono-allelic loss. This result indicates that other gene(s) located on 6q are of importance
in melanomagenesis. As suggested by the referee we have added to our manuscript the ³ reference where
they analyzed 53 cases.

1Q6.

2) Although the authors convincingly show with their mouse model that partial loss of BRN2 increases
melanoma proliferation, this finding is expected from many previous publications, as the authors already
discuss. The authors go on to claim that haplo-insufficiency is "sufficient" for both initiation and
metastasis. However, these conclusions are not substantiated by the presented data:

1A6.

1) We do not claim that haplo-insufficiency of Brn2 is "sufficient" for both initiation and metastasis. We
claim that the haplo-insufficiency of Brn2 promotes more efficiently both initiation and metastasis
formation on a (Tyr::CreERT2^o; Braf oncflox/+; Pten f/+) genetic background that was tamoxifen induced.
It was very clear in the legend and the text. Perhaps, it was confusing in the abstract. We changed this
sentence as follows "Here, in a BrafV600E Pten+/- context, we show that BRN2 haplo-insufficiency
promotes melanoma initiation and metastasis."

2) The referee focuses only on proliferation (Figure 2C). There is another very important issue to consider;
the number of independent melanomas is higher in Braf-Pten-Brn2-het/hom mice than in Braf-Pten-Brn2-
WT mice showing that independent initiation is promoted when the Brn2 gene is defloxed (Figure 2B). We
make a point on this issue in the discussion.

1Q7.

a. In the manner the experiments were conducted, the perceived increase in both initiation and
metastasis could be explained by increased proliferation. For example, if the melanocytes proliferated
faster (as the authors demonstrate) than at any given time point, the number of measurable tumors will

be greater, even if the number of initiating events is the same. The authors should consider comparing
the number of tumors per mouse at different time points for each background, when the average tumor
size is the same.

1A7.

The referee makes an excellent point and in the revised version we have tried to be more precise in the
description of the experiments.

Braf-Pten-Brn2-het mice were sacrificed on average **1.3 weeks** after appearance of the first tumors with
**an average of 16 tumors/mouse**. Similar results were obtained with Braf-Pten-Brn2-hom mice. Braf-Pten-
Brn2-WT mice were sacrificed at **4 weeks with an average of 8 tumors** even they did not reach a total
volume of 2cm³ except one mouse that was sacrificed earlier (three weeks).

As mentioned in the Materials and Methods, mice were killed either four weeks after the appearance of
the tumors or when the total volume of the tumors reached 2 cm³. We added in the materials and
methods that the mice were checked two-three times a week to estimate the volume of the tumors. The
given estimation was performed around the time of sacrifice.

These results collectively support the conclusion that initiation (proliferation and senescence
[bypass/escape]) is more effective when the amount of Brn2 is lower than normal (het or hom). We
added this information in the figure legends.

1Q8.

b. Similarly, the authors claim an observed increase in metastasis. However, the authors note that the
number of metastatic events were actually the same. This is despite an increase in primary tumor number
and size. The observed differences were the size of the lymph node and the number of melanocytes in the
node – both of which could be explained by more rapidly dividing cells. A more reasonable conclusion is
that the number of metastatic events is the same upon BRN2 loss (or even less relative to total primaries),
but that the BRN2 het melanocytes proliferate faster. The authors should consider quantifying distal
metastases, which the activated BRAF, PTEN deleted mice are known to induce. If partial loss of BRN2 can
substitute for the loss of the second PTEN allele, the expectation is that activated BRAF, partial PTEN loss,
partial BRN2 loss mice will also present with distal metastases.

1A8.

Our goal was not to say that the “number of metastatic events were actually the same” as the referee
wrote. Indeed, we observed bigger lymph nodes and more metastatic pigmented cells in Braf-Pten-Brn2-
het LN than in Braf-Pten-Brn2-WT LN and Braf-Pten-Brn2-hom. In vivo, we may have more metastatic cells
because we have more primary tumors, more proliferation or/and more migration/invasion abilities. The
amount of metastasis is higher in the case of Braf-Pten-Brn2-het (het) melanoma cells compared with
Braf-Pten-Brn2-WT (WT) and Braf-Pten-Brn2-hom (hom) melanoma cells (Figure 4A-C). According to our
results, the number of primary tumors and the rate of proliferation are higher for het and hom melanoma
cells, and according to Zeng et al (2018) the migration of hom melanoma cells would be handicapped
compared to het and WT melanoma cells. On one hand, het and hom melanoma proliferate faster (Figure
3A-D) and on the other hand, hom melanoma are exhibiting reduced invasion (according to our results
(Figure 4F,G), migration⁴ and death by apoptosis and/or anoikis^{5,6}. Altogether, one may understand that
melanoma grow faster and form melanoma when the level of Brn2 is not too high (proliferation handicap)
but not too low (migration/invasion/survival handicap).

We have added a sentence in the discussion on this issue.

In our conditions, we did not observe distal metastases during the follow-up period that is constrained by
ethical and regulatory guidelines. Concerning the suggestion to quantify distal metastases, several BRAF
PTEN melanoma models exist and are not fully identical (there are two Tyr::CreERT2 mouse models
(Bosenberg and Larue) and two Brafonc model (McMahon and Marais). Only three mouse melanoma
models were generated – the Tyr::CreER^{T2} Bos mice were never crossed with the Brafonc Marais model as
far as we know. According to their exact genotypes, genetic background and animal colony, mice may
present or not distal metastasis. In our conditions (Tyr::CreER^{T2} Lar, and Braf onc Marais), we never
observed any distal metastasis besides LN.

1Q9.
c. The authors only observe a difference on the specific background of activated BRAF and PTEN partial
loss, but neither PTEN diploid nor PTEN full loss. This is a niche circumstance, the relevance of which to
human melanoma is unclear, as PTEN loss generally occurs quite late during human melanoma
progression. In either case, the conclusions of BRN2 loss being “sufficient” for any of these phenotypes
are misleading. The authors also chose not to present the PTEN full loss data, nor comment on BRN2 full
loss data, both of which are critical controls.
1A9.
We agree with the reviewer that the BRAF activated and a PTEN partial loss represents a specific genetic
background. These two genetic events (BRAF and PTEN) are the most frequent alterations of the MAPK
(50-60%) and PI3K (30-50%) signaling pathways, respectively. The activation of BRAF and the loss of PTEN
are two independent events occurring in about 15-30% of the cases. In this sense, we do not believe that
it is a niche circumstance for these patients. We developed this point for humans in Figure S2.
We now included the Pten diploid/full loss and Brn2 full loss:
(i) Comment on Pten diploid (WT).
Indeed Brn2 regulates positively the transcription of Pten. In the absence of Brn2, Pten may be down
regulated but it is not sufficient to act efficiently on the tumor-suppressor function of Pten. We included
this point in the text.
(ii) Comment on Pten loss
In the absence of PTEN on a Braf^{V600E} background, the appearance of the tumors is too rapid to observe
any difference between Brn2+/+, Brn2 f/+ or Brn2 f/f. According to our model the transcription regulation
by Brn2 and Mitf would not affect the level of Pten since it is already lost. This information is now
included in the text.
(iii) Comment on BRN2 full loss
The number of tumors/mouse and growth of the Braf-Pten-Brn2-hom tumors are now presented in Figure
2. The number of tumors/mouse and the growth rate of the tumors are similar in Braf-Pten-Brn2-het and
Braf-Pten-Brn2-hom mice. The appearance of the tumors in Braf-Pten-Brn2-hom and Braf-Pten-Brn2-het
is similar (Figure S3). This information is now included in the text and the full data are now presented.
1Q10.
3) The data to support the claim of two novel mechanisms linking BRN2 loss to PTEN expression are
preliminary.
1A10.
Regarding this point, concerning the regulation of PTEN by BRN2, we show that:
(i) In vivo the level of PTEN is reduced when Brn2 is reduced.
(ii) In cellulo we show that the reduction of Brn2 induces a reduction of Pten mRNA in three independen
human melanoma cell lines (DAUV-1, Gerlach and SKmel28).
(iii) ChIP experiments shows that Brn2 binds to the promoter and to exon 1 of Pten.
(iv) Initially, we contacted several times a group at the MD Anderson to get the human PTEN promoter,
but were unfortunately unable to obtain it from the authors of BBRC, 292, 422-426. “Promoter analysis of
tumor suppressor gene PTEN: identification of minimum promoter region”. We therefore cloned the PTEN
promoter upstream of a luciferase reporter and showed that increases expression of BRN2 activates the
PTEN promoter and decreasing the BRN2 level decreases PTEN promoter activity. It was indeed a real tour
de force to get this construct since the amount of GC is very high in this promoter. In order to show that
this regulation is conserved during evolution, we also cloned the mouse PTEN promoter. We showed that
BRN2 regulates PTEN promoters in both species. These constructs will of course be available for the
community.

The second mechanism is the repression of PTEN transcription by Mitf, we show that
(i) In cellulo the reduction of MITF induces an up regulation of Pten mRNA and protein in three human melanoma cell lines (HBL, Mel501 and SKmel28), but not in DAUV-1, nor Gerlach since the level of mRNA of Mitf was already too low.
(ii) Cut and Run seq experiments were performed in SKmel28 melanoma cell lines deleted or not for MITF (SK28 ΔX6). These show that Mitf may bind on intron 4 and at +114 kb. In addition to these sites, we validated a third binding site of Mitf on Pten (around +140kb) by a classical ChIP analysis using the 501Mel melanoma cell line.
(iii) Using the same Cut and Run seq approach, four distal enhancers exhibited greater (at least 2-fold) H3K27ac signal in MITF mutant cell lines compared to wild-type cell lines (+140kb, +210kb, +230kb, +253kb), suggesting increased transcription of PTEN in the former.

We believe that at this point, these data are sufficient to show that BRN2 and MITF regulate the transcription of PTEN.

1Q11.

a. The microarray results provided are under-powered. In the methods the authors state that 13 tumors were profiled. In the presented results, only 7 total were used, inclusive of 5 experimental and 2 controls. What was the rationale for omitting the other 6 tumors? Were the samples representative of independent mice? Why were only 2 controls used? That the study was under-powered is clear from Table S3 which shows that no genes were significantly altered in expression when considering the adjusted p value.

1A11.

The reviewer is correct. We increased the number of samples to generate similar information. Indeed we initially wanted to profile 13 tumors but we had a major problem with the extraction of RNA from WT for unknown reasons. In this respect, we generated more mice and tumors from new crosses. We produced five more Braf-Pten-Brn2-het tumors and three more Braf-Pten-Brn2-WT tumors. We also generated tumors that were homozygous for Brn2 (n=10). RNAs were extracted prior to repeating the transcriptomic analysis with all samples (10 from Braf-Pten-Brn2-het, and 5 from Braf-Pten-Brn2-WT tumors, and 10 from Braf-Pten-Brn2-homo). A new Table S3 is now provided. Moreover, we indicated in the Materials and Methods that all tumors were derived from independent mice. We analyzed these transcriptomic analyses accordingly. The results are given in Figure 4 and Figure S5. The main conclusions of this novel analysis are given in the text. Mouse melanoma tumors and mouse melanoma cell lines microarrays have been deposited on Gene Expression Omnibus (GEO) website under accession number GSE126524 and GSE163085, respectively. A SuperSeries identifier including both cell lines and tumors is GSE163086.

1Q12.

b. The relationships between either BRN2 or MITF and PTEN are both novel and of interest to the community. These studies could be strengthened by performing the experiments in more than one human cell line, by using a luciferase reporter to determine whether transcription factor binding increases or decreases transcription, and by performing both the BRN2 and MITF experiments in the same sets of cells. As is, it is unclear how the MITF data presented in figure S8 are connected with the rest of the manuscript.

1A12.

We partly answered this question in 1A10.

* First, it is important to estimate the amount of mRNA (PTEN, BRN2 and MITF) in the four human melanoma cell lines (Gerlach, Dauv-1, SK28, and 501Mel) that we used. These cell lines produced very high amount of PTEN (20<Ct<22), high amount of BRN2 (22<Ct<24), and highly different amounts of MITF (19<Ct<26). 501Mel cells had the lowest level of BRN2 (Ct=24) and the highest level of MITF (Ct=19). Dauv-1 and Gerlach had the lowest levels of MITF (Figure S6).

* We initially evaluated the connection between BRN2 and PTEN on two cell lines. One was human (Dauv-1) and one is mouse (melan-a). It shows that this regulation is conserved during evolution; this is an important point to stress. We tested our hypothesis on three novel human melanoma cell lines (Gerlach SKmel28, and 501Mel) as requested by the reviewer. As requested, we performed the knock down of BRN2 in these three cell lines (Gerlach, SK28, and 501Mel), and we confirmed that the reduction of BRN2 reduces the level of PTEN in cells having a high level of BRN2 (three human cell lines and in one mouse cell line), but not in cells (501mel) having a lower level of BRN2 (see Figure 5 and Figure S6).

* We initially evaluated the connection between MITF and PTEN on three cell lines (501Mel, HBL and SK28). Now, we performed the knock down of MITF in Gerlach and Dauv-1 cells, we observed an induction of PTEN only in cells having a high level of MITF (e.g., 501Mel, SK28), but not in cells having a lower amount of MITF such as Gerlach and Dauv-1 cell lines (Figure S6). From the original figure S8, similar results were obtained with two other cell lines. These results are shown in Figure S6 (si BRN2 and si MITF) and Figure 6 (si MITF).

In the discussion, we come back on the high complexity of regulation of MITF by BRN2 associated with AXL.

To answer the referee concerning the luciferase reporter experiment, we cloned, sequenced and generated the mouse and human PTEN promoter constructs driving luciferase. We transfected the human PTEN-luciferase and the mouse Pten luciferase constructs in the DAUV-1 and SK28 human melanoma cell lines and three mouse melanoma cell lines (m50, m59 and m82) that we have established from the different tumors (Braf-Pten-Brn2-WT, Braf-Pten-Brn2-het, and Braf-Pten-Brn2-hom), respectively (Figure 5 and figure S7). As expected, the knockdown or the exogenous expression of BRN2 reduced or induced the PTEN promoter activity in all these cells, except in the m82 cells that do not produce BRN2 since they are derived from a BRAF-PTEN-BRN2-hom tumor.

* Figure S8 on MITF is now a main figure (Figure 6), since we brought additional information on the regulation of PTEN by MITF.

1Q13.

4) The title, abstract, introduction and discussion contain misrepresentations of the previous literature and exaggerations of the results.

a. Although the authors cite the majority of relevant work, this reviewer disagrees with the representations of the data within those studies and suggests a more careful reading and comprehensive assessment of the cited papers. Specifically, the contribution of BRN2 to melanoma metastasis has been looked at in vivo by multiple groups as has the consequence of BRN2 inhibition in normal melanocytes, and the relationship between MITF and BRN2 is more complex than suggested.

1A13.

We would like to answer this point, but it would have helped if the referee had been more precise. We have added several citations that we omitted in the previous version.

The referee considers that the contribution of BRN2 to melanoma metastasis has been looked at in vivo. If we consider that xenografts experiments are in vivo experiments, one may agree with the referee.

However, these experiments are not fully 'in vivo' since the cells were grown in culture on plastic in the presence of serum calf serum, etc.... and were then engrafted into immune deficient mice. Thus the experiments may be in vivo, but do not accurately reflect the physiopathological disease, especially regarding the immune system that may affect the ability of cells to metastasize. In this respect, the conclusions of the previous studies should be considered as important, but not fully reflecting the in vivo situation faced by melanoma cells in humans. Nevertheless, we modified the text accordingly but

maintain is a fact that no laboratory has evaluated the role of Brn2 during melanoma initiation before this current study.

We fully agree with the reviewer that the relationship between MITF and BRN2 is highly complex. Here the goal of this article is not to study the relationship between MITF and BRN2, but it is to bring to the light for the first time that PTEN plays a role in this complexity.

1Q14.

b. Many of the conclusions drawn are exaggerated and need to be toned down. In particular, conclusions based upon the TCGA analyses and microarray data should be omitted unless the issues outlined above are addressed.

1A14.

We have now moderated our conclusions when needed. All TCGA and microarray data studies are correlative. In this respect, we cannot conclude until appropriate functional studies are performed.

Reviewer #2 (Remarks to the Author):

In this study the authors argue that BRN2 is a haplo-insufficient tumor suppressor whose loss promotes melanoma initiation and progression. The mechanistic basis for this is proposed to be BRN2 regulation of PTEN transcription, either directly or via repression of MITF. Whereas previous studies have indicated that BRN2 promotes melanoma invasion (e.g. Goodall et al., 2008; Fane et al., 2017), the role of BRN2 in melanoma initiation and progression remains to be fully investigated. This report provides important clues and contributes to a better understanding of the role of this transcription factor in melanomagenesis and possibly other BRN2- dependent tumors

The BRN2 locus shows frequent copy number loss in cutaneous melanoma metastases, although these deletions also affect many other genes implicated in melanomagenesis (Fig. 1A-B). Additional data supporting that BRN2 is the key contributor to melanoma initiation and progression would significantly strengthen the manuscript.

Major issues

2Q1

Mono-allelic loss of BRN2 is associated with worse overall survival. Considering that central thesis is that BRN2 is a tumor suppressor whose monoallelic loss promotes melanoma initiation and progression, frequency of BRN2 loss (in addition to mRNA levels) at early/pre-metastatic stages of melanoma should be included to further support the weak association BRN2 low vs. high mRNA levels with survival (Fig. 1G). Is this an early event in melanomagenesis?

2A1.

By definition, in our functional assay, the activation of Braf and the inactivation of Pten and Brn2 occur at the same time. Of course it is different in humans where the various genetics, epigenetics and allogeneics modifications (may) occur at different times and in any order. The referee would like us to provide more information on the evolution of the expression of BRN2 during the natural history of melanoma.

We performed the analysis of the publically available data published by Shain et al (2018) to evaluate the number of BRN2 alleles in nevi and melanoma that arose from these nevi. Nineteen pairs (nevus/melanoma) could be analyzed as shown in Figure S1D. It appears that 25% (5 out of 19) of the melanoma presented a mono-allelic loss of BRN2 compared to nevi. In this respect, we may conclude that this BRN2 mono-allelic loss occurred during the early steps of melanomagenesis.

2Q2.

Mouse model indicates that mono-allelic loss of BRN2 in neonatal mice promotes melanomagenesis. Recommend further experiments to support BRN2 tumor suppressive function.

2A2.

Being on a pure C57BL/6 background (crucial), this mouse model demonstrates, and not only indicates, that mono-allelic loss of BRN2 in neonatal mice promotes melanomagenesis in a BRAF-PTEN background (Tyr::CreERT2⁺; Braf oncflox/+ ; Pten f/+).

The understanding of tumor initiation cannot be performed *in vitro* since the cells/melanocytes (i) are established in culture in a non physiological environment, (ii) are adapted to a novel environment (including plastic, serum, trypsin), which means that there are modifications of gene expression (including transcription, splicing, translation, post translation modifications and mRNA/protein degradation) from DNA to active protein, (iii) carry various mutations and (iv) are very often immortalized in vitro. Today, it is almost impossible to work with primary culture to address such question. In this respect, the only way to

448 show the causal role of a gene/protein during melanoma/tumor initiation is to use genetically engineered
animal model (GEM).

Of course, to decipher the various cellular mechanisms occurring during melanomagenesis, it is indeed
crucial to perform in cellulo experiments with the associated limitations. Our study demonstrates that
Brn2 acts as a tumor-suppressor, this was not shown previously, and as pointed out above is not only
relevant to the genetics of melanoma, but also to conditions when BRN2 might be down-regulated by
conditions in that microenvironment that would decrease its promoter activity.

2Q3.

Does re-expression of BRN2 in cells that have lost BRN2 suppress (i) colony formation in vitro?

2A3.

* Colony formation represents the ability of the cells to grow and to form a colony in vitro which is an
assay to evaluate some parts of the metastasis process. Colony formation is associated with melanoma
progression but not with melanoma initiation.

* In two different studies it was shown that the reduction of Brn2 levels has no effect on colony formation
(Simmons et al.2017 and Herbert et al., 2019). In Simmons et al, 2017, they show that “Brn2 knockdown
cells were able to efficiently form colonies to the same level as the cellular population expressing control
shNEG”. In Herbert et al, 2019, it was shown that BRN2 depletion using 2 different siRNA do not change
the clonogenic survival of cells compared to the siControl.

Nevertheless, we addressed the referee’s question and performed a series of colony formation
experiments in which we reduced the level of Brn2, and re-expressed Brn2, in appropriate mouse
melanoma cell lines that we established from our mouse melanoma models BRAF;PTEN;BRN2. BRN2 was
either +/+, F/+ or F/F. All these cell lines are on a C57BL/6 background and are mutated for BRAF and
PTEN. We evaluated the capacity of these cell lines to form colonies (CFU) (Figure S5G-I). In this case, the
number of CFUs is independent of the presence or absence of Brn2. Moreover, we infected with a Brn2
expression vector two independent mouse melanoma cell lines that lacked Brn2 (m8 and m82). Again, it
appears that the re-expression of Brn2 in these cell lines does not affect their capacity to form colonies.
All these information are in agreement with the previous published results in human cells and are now
included. Moreover, it confirms that the murine melanoma model used generates data similar to human
cells.

2Q4.

Does re-expression of BRN2 in cells that have lost BRN2 suppress **(ii) tumor formation in mice?**

2A4.

The presence or absence of Brn2 did not decrease the ability of these melanoma cell lines to grow in
syngeneic mice (data not shown). In other words, it appears that the absence of Brn2 in these melanoma
cells does not affect the implantation of the cells on the body wall, the proliferation after their
transformation or the induction of angiogenesis in an immunocompetent environment. We included this
information in the text.

2Q5.

Would be useful to show survival data for the different cohorts of mice as well as histology of the primary
tumors including staining with melanoma and proliferation markers.

2A5.

1) We included the Kaplan-Meier curve for the appearance of the first tumor for Braf-Brn2 (+/+, f/+ and
f/f) and Braf-Pten-Brn2 (+/+, f/+ and f/f) mice as supplemental data. In the case of mice the survival is
artificial since we needed, for ethical reasons, to sacrifice the mice when the tumors reached a total
volume 2cm³ or four weeks after appearance of the first tumors. This information is now given in the text

(Materials and Methods for the criteria and information in the figure legend) (Figures S3).

2) The Braf-Pten melanoma models are widely used in the field and they are accepted to be one of the
melanoma models of choice. See for instance Dankort et al (2009); Damsky et al (2011); Scotegegna et al
(2014), Rodriguez et al (2018), Laurette et al (2019).

3) In Figure 3A-C, we already presented BrdU and Ki-67 staining of the melanomas and showed that the
proportion of Ki-67+ cells was higher in Braf-Pten-Brn2 mouse melanoma than in Braf-Pten mouse
melanoma. Similar results were obtained for the BrdU experiments.

2Q6.

Do tumors progress to metastatic melanoma beyond LN infiltration?

2A6.

The growth of the tumors is so rapid that we had to sacrifice the mice for ethical issues. Consequently a
full analysis of metastasis cannot be performed with these transgenic mice according to ethical rules.

2Q7.

What effects does bi-allelic loss of BRN2 in melanoma cells have on tumorigenicity?

2A7.

We now included the data corresponding to melanoma initiation in Brn2 homozygous mice (Figure 2). On
this genetic background (Braf-Pten), the appearance of melanoma is similar when Brn2 is homozygous or
heterozygous. However, the formation of metastasis is reduced in Braf-Pten-Brn2-hom than in Braf-Pten-
Brn2-het. All these information are included in the text with the corresponding figures. The results
therefore suggest that a certain level of BRN2 is required for successful metastatic colonization. One
possible explanation is that BRN2 can suppress anoikis⁶, a process of cell death that is associated with the
metastatic cascade. On the top of this, a series of results, including ours, show that the presence of BRN2
is required for invasion^{4,6-8}

2Q8.

What is the impact of silencing the other genes in Chr6 (e.g. Arid1A, ROS)?

2A8.

While we focused our study on Brn2, we regard this as an important question.

Similar in vivo experiments could be performed with Ros and/or Arid1a mice. Knowing that Arid1b may
interact with Brg1 (Smarca4), we may expect to have a positive output according to our results obtained
previously⁹. Arid1b mice homozygous for a null allele die perinatally. Arid1b Floxed mice (exon 5) were
generated¹⁰. The full knockout of Ros1 was performed by Carmen Birchmeier¹¹. In the absence of Ros1,
mice are viable but males are not fertile, it is therefore possible to address this question with such mice.
To generate such combination (CreERT2/Braf/Pten/Arid1A or Ros) on a pure genetic background, produce
enough mice, follow the tumors, it would take over than 3 yrs. These experiments cannot reasonably be
expected from us in a reasonable time. Moreover, this article provides a large amount of novel
information on the function of BRN2.

Since the levels of Arid1b and Ros in mouse melanoma lacking or not Brn2 were not affected, the
phenotype arising as a consequence of loss of Brn2 and is not obviously link to Arid1B/ROS levels in the
mouse. This information was in the discussion (p17 of the previous ms). In addition, we showed that the
level of expression of Arid1b or Ros are not correlated with the OS in patients (see Figure S1)

2Q9.

Evidence from clinical samples of melanoma in situ/invasive melanoma vs. adjacent precursor nevi
suggests that BRN2 levels increase during transition to invasive melanoma (Zeng et al., 2018). Please

discuss this and address whether mono-allelic loss of BRN2 occurs early in melanoma development.

2A9.

In the Zeng model, they observe that the loss of CDKN2A locus leads to the upregulation of CDK4/6 that
induces E2F1, which is able to induce BRN2 and promoting in their assay motility and invasion. This
observation is not incompatible with our observation even though we are not in the same context; the
CDKN2A locus is intact.

In our model, the level of Brn2 may increase temporally since the mice are heterozygous for Brn2, by
classical induction of Brn2 gene expression. Since Brn2 heterozygosity induces metastasis but Brn2
homozygosity does not, it fits perfectly with the Zeng model. We consequently added a sentence in the
discussion.

Altogether, the rate of proliferation and the number of melanomas are higher in Braf-Pten-Brn2-het (het)
and Braf-Pten-Brn2-hom (hom) mice than Pten-Pten-Brn2-WT (WT) mice, but are similar between het and
hom mice. However, there are more metastasis in het mice than in hom mice. This is that is consistent
with the Brn2 expression being required for invasion and metastasis (see Figures 2-4).

2Q10.

All main figure in vitro analyses rely on comparisons of one transformed cell line and one non-
transformed melanocyte (e.g. following BRN2 knockdown; Fig. 6D). For greater generalizability please
include additional cell lines such as those in Fig. S8 and include cells with loss of PTEN.

2A10.

We acted accordingly and added more cell lines

* First, we estimated the amount of mRNA (PTEN, BRN2 and MITF) in the four human melanoma cell lines
(Gerlach, Dauv-1, SK28, and 501Mel) that we used. These cell lines produced very high amounts of PTEN
(20<Ct<22), high amounts of BRN2 (22<Ct<24), and highly different amounts of MITF (19<Ct<26). 501Mel
cells had the lowest level of BRN2 (Ct=24) and the highest level of MITF (Ct=19). Dauv-1 and Gerlach had
the lowest levels of MITF (Figure S6).

* We initially evaluated the connection between BRN2 and PTEN in two cell lines. One was human (Dauv-
1) and one is mouse (melan-a). It shows that this regulation is conserved during evolution; this is an
important point to stress. We tested our hypothesis on three novel human melanoma cell lines (Gerlach
SK28, and 501Mel) as requested by the reviewer. As requested, we also performed the knock down of
BRN2 in these three cell lines (Gerlach, SK28, and 501Mel), and confirmed that the reduction of BRN2
reduces the level of PTEN in cells having a high level of BRN2 (three human cell lines and in one mouse
cell line), but not in cells (501mel) having a lower level of BRN2 (see Figure 6 and Figure 6S).

* We initially evaluated the connection between MITF and PTEN on three cell lines (501Mel, HBL and
SK28). We performed the knocked down of MITF in Gerlach and Dauv-1 cells, we observed an induction of
PTEN only in cells having a high level of MITF (e.g., 501Mel, SK28), but not in cells having a lower amount
of MITF such as Gerlach and Dauv-1 cell lines (Figure 6S). From the original figure S8, similar results were
obtained with two other cell lines. These results are shown in Figure 6S (si BRN2 and si MITF) +Figure 6 (si
MITF).

In the results and discussion, we are coming back on the high complexity of regulation of MITF by BRN2
that includes their mutual regulation. Moreover, we bring to the light AXL and its regulation. Indeed, AXL
is associated with melanoma metastasis. We have shown that Braf-Pten-Brn2-het cells produce more Axl
than Brn2-WT/hom (see Figure 4H). Moreover, when the levels of both Brn2 and Mitf are decreased in
human melanoma cell lines, the level of AXL is induced (see Figure S6D,H,L,P,T).

* Figure S8 on MITF is now a main figure (Figure 6), since we brought additional information on the
regulation of PTEN by MITF.

2Q11
Authors suggest that BRN2 may induce PTEN transcription partially by repressing MITF. While references
for the link between BRN2 and MITF are cited, no evidence is presented to show that BRN2 inhibits MITF
expression in these models or that BRN2 knockdown/loss enhances MITF expression. PTEN levels are
instead monitored following direct knockdown of MITF. Include data showing regulation of MITF by BRN2
and corresponding PTEN levels.

2A11
The reviewer is right, we do not present any evidence that BRN2 inhibits MITF expression in this mouse
model. As previously showed in Figure 1 from ⁹, the BRAF/PTEN model is pigmented only early on. On day
29 after tamoxifen treatment, there are still some pigmented cells but the majority of the cells are
unpigmented. This observation is perfectly logical according to our transcriptomic analysis in which we
show that the level of M-Mitf is very low. Any repression of Mitf by Brn2 would therefore be transient
during the early process of transformation.

In the literature, it has been shown that depletion of endogenous BRN2 from different melanoma cell
lines (A375, WM266-4, Colo829, SKMel-28) can result in the reduction of the level of MITF to various
extents, but not in MM649, A02 or MM455 cells ^{8,12,13}. The expression of exogenous BRN2 leads to the
induction of MITF in MM455 cells (Mitf low), reduces the level of MITF in MM603 cells (Mitf high), and
does not affect the level of MITF in MM370 cells (Mitf very high) ⁶. Finally, induction of BRN2 may also
lead to the reduction of MITF expression in 501mel cells ⁷. Using an M-Mitf promoter driving luciferase
construct, the level of luciferase was reduced after expression of BRN2 in mouse melanocyte (melan-a)
and human melanoma (Lyse) cell lines ¹⁴. All these experiments show that the regulation of Mitf by Brn2 is
complex. It has to be pointed out that all these experiments were not performed the same way.

We performed some in vitro experiments with four human melanoma cell lines (Gerlach, DAUV-1, SK28
and 501Mel) to evaluate the consequences of the transient reduction of Brn2 and/or Mitf on the level of
Pten mRNA (Figure S6). Aside from this, we evaluated the consequences of these reductions on BRN2,
MITF and AXL. In our experimental conditions, we do not observe any effect of the reduction of Brn2 on
the level of Mitf mRNA (Figure S6). We performed the same experiments as we did in Berlin et al ¹⁴ with
501Mel cells line and showed that the level of Mitf-luciferase was reduced after expression of BRN2.

2Q12.
To demonstrate that BRN2 loss can also repress PTEN indirectly via the induction of MITF, concomitant
silencing of BRN2 and MITF should be performed. Likewise, MITF levels could be assessed in
BRAF/PTEN/BRN2 tumors.

2A12.
We showed that Brn2 can bind directly to the Pten promoter, and that Brn2 induces the Pten promoter
using a classical luciferase assay. We also show that independently Mitf represses Pten mRNA levels. See
Figures 5 and 6 and associated supplemental. Both events may occur during the process of
melanomagenesis and may counteract or not according to the fact that Brn2 induces or represses Mitf
according to the cellular environment. We performed a concomitant silencing of BRN2 and MITF in the
four human melanoma cell lines. In our experimental conditions, siBRN2 counteracts siMITF in the
regulation of PTEN. As already mentioned previously, Mitf is not expressed in late BRAF/PTEN/BRN2
mouse melanoma. It is now shown in Figure S6.

2Q13.
MITF promotes proliferation, whereas low MITF is associated with an invasive, slow-cycling phenotype.
Could MITF also be contributing to the effects of Brn2 loss on proliferation and LN infiltration?

2A13.
The information given by the referee "MITF promotes proliferation, whereas low MITF is associated with
an invasive, slow-cycling phenotype" is coming from some melanoma cells in culture and are accepted as

dogma.
1) Is it true for all melanoma cell lines in cultures? NO. Some human melanoma cell lines (e.g., Lu1205)
lack Mitf, they do proliferate very nicely, and the reintroduction of Mitf in these cells does not promote
proliferation. Are there molecular compensations? Nobody addressed this question to my knowledge.
2) Is it fully true in vivo? The question raised by the referee is of great importance. What is the causal role
of MITF in melanomagenesis? Today, nobody answered directly this question in vivo in mammalian
melanomagenesis. In order to answer this question, we have to knock out MITF once the melanocytes are
established after embryogenesis in vivo in a physiopathological melanoma model. Such experiment will
have to be performed in the future as a full project in the presence and absence of Brn2.

2Q14.
Since the overall conclusion of the study is that “BRN2 loss reduces PTEN transcription in vitro and in vivo,
thus ramping up PI3K signaling and inducing both the initiation of melanoma and the formation of
metastases” it would be interesting to determine the effects of BRAFi+PI3Ki inhibitors on
BRAF/PTEN/BRN2 tumor initiation and LN metastasis.

2A14.
From the established mouse melanoma cell lines we evaluated the action of BRAF V600 (PLX4720), MEK
(Binimetinib), and PI3K (LY294002) inhibitors and determined the IC50 of these drugs after CFU assays.
Brn2-het and -hom cells are more sensitive than WT cells. We included this information as figure S4. It has
been previously showed by Herbert and colleagues (2019) that depletion of BRN2 sensitised the cells to
BRAF inhibitors. Such experiments were not performed genetically and other drugs were not used. As
requested by the referee, we evaluated the cooperation of PLX4720 and LY294002 using various
concentrations of each drug, but we never observed any cooperation/synergy of these two drugs using
CFU assay.

Though undoubtedly a good experiment, our research ethics committee did not approve the use of
BRAFi+PI3Ki inhibitors in mice before the tumors appear. Therefore we could not assess their effects on
melanoma initiation.

Minor issues

2Q15.
BRN2 mRNA levels are lower in invasive melanomas (T3+T4) than localized. Please incorporate Fig. S2B
into Fig. 1 as this shows the most clinical relevant staging (Clark’s level superseded by AJCC staging).
What is PTEN mutation status in cell lines used?

2A15.
T1 and T2 are already invasive melanoma by definition (AJCC classification). We are unsure of the
meaning of “localized”; when it is limited to epidermis it is referred as in situ (Tis). However, there is a
conflict with the request from reviewer #1. In order to avoid any misrepresentation we removed the
former Figure S2B that is not essential for the main message of the article.

We included a table (Table S3) giving the genetic status for BRAF and NRAS, and the relative protein level
of BRN2 and PTEN of the melanoma cell lines used in this study.

2Q16.
Based on the phenotype elicited by loss of Brn2, the authors state that “Brn2 acts as a tumor suppressor
in vivo, and its loss induces melanoma initiation and increases tumor growth rate”; reword to more
precisely indicate that loss of Brn2 in the context of monoallelic PTEN...

2A16.
We modified the sentence. Indeed, we tested in mice the role of Brn2 in a context Braf(V600E) and Pten
heterozygous and showed that Brn2 acts as a tumor suppressor.

2Q17.
"Cell lines express Pten and Brn2 mRNA...(data not shown)" – include the data in supplemental figures.
2A17.
It is now included as Figure S6 and table S3.
2Q18.
"Cell number was monitored" after BRN2 knockdown (Fig. 4D). Cell cycle analysis to assess relative
proportion of cells proliferating would further support the conclusions.
2A18.
The number of BrdU+ cells is determined from the solid tumors (Figure 3C,D), as Ki-67 (Figure 3A,B). We
tried to perform a FACS analysis of these tumors to better evaluate the different phases of the cell cycle
but we were not successful.
2Q19.
Transcriptomic analysis of tumors from BRAF-PTEN-BRN2 vs. BRAF-PTEN-WT (Fig. 5). Please indicate
selection criteria for which tumors were analyzed. Was tumor size matched?
2A19.
Only one tumor per mouse was considered corresponding to the biggest one.
Tumors had the same size since we harvested tumors for transcriptomic analyses when they reached a
size of 1 cm³. This information is now included in the materials and methods.
2Q20.
Reduction in PTEN positivity upon BRN2 loss (mouse tumor IHC – Fig. 6A) is not reflected in Western blot
levels (Fig. 6B). PTEN protein levels look comparable between WT and BRN2 when normalized to actin
intensity. Provide densitometry values normalized to actin in Fig. 6B.
2A20.
We disagree. The reduction of Pten protein after the reduction of the level of Brn2 is reflected in the
western blot analysis. The intensities of the bands were determined and the values are now shown
(Figure 5B). Similar results were obtained with different tumors (at least three of each).
2Q21.
Also probing for MITF is suggested.
2A21.
Mitf is expressed at a very low level in advanced Braf/Pten tumors (see Laurette et al, 2020 – Figure S3).
Unfortunately, we could not detect MITF neither by western blotting nor by IHC in these mouse tumors.
**References for both referees**
1. Bastian, B.C., LeBoit, P.E., Hamm, H., Brocker, E.B. & Pinkel, D. Chromosomal gains and losses in
primary cutaneous melanomas detected by comparative genomic hybridization. *Cancer Res* **58**, 2170-5
(1998).
2. Guan, X.Y. *et al.* Detection of chromosome 6 abnormalities in melanoma cell lines by chromosome arm
painting probes. *Cancer Genet Cytogenet* **107**, 89-92 (1998).
3. Healy, E. *et al.* Prognostic significance of allelic losses in primary melanoma. *Oncogene* **16**, 2213-8
(1998).

- 4. Zeng, H. *et al.* Bi-allelic Loss of CDKN2A Initiates Melanoma Invasion via BRN2 Activation. *Cancer Cell*
**34**, 56-68 e9 (2018).
- 5. Herbert, K. *et al.* BRN2 suppresses apoptosis, reprograms DNA damage repair, and is associated with a
high somatic mutation burden in melanoma. *Genes Dev* **33**, 310-332 (2019).
- 6. Pierce, C.J. *et al.* BRN2 expression increases anoikis resistance in melanoma. *Oncogenesis* **9**, 64 (2020).
- 7. Goodall, J. *et al.* Brn-2 represses microphthalmia-associated transcription factor expression and marks
a distinct subpopulation of microphthalmia-associated transcription factor-negative melanoma cells.
*Cancer Res* **68**, 7788-94 (2008).
- 8. Simmons, J.L., Pierce, C.J., Al-Ejeh, F. & Boyle, G.M. MITF and BRN2 contribute to metastatic growth
after dissemination of melanoma. *Sci Rep* **7**, 10909 (2017).
- 9. Laurette, P. *et al.* Chromatin remodellers Brg1 and Bptf are required for normal gene expression and
progression of oncogenic Braf-driven mouse melanoma. *Cell Death Differ* **27**, 29-43 (2020).
- 10. Celen, C. *et al.* Arid1b haploinsufficient mice reveal neuropsychiatric phenotypes and reversible causes
of growth impairment. *Elife* **6**:e25730(2017).
- 11. Sonnenberg-Riethmacher, E., Walter, B., Riethmacher, D., Gödecke, S. & Birchmeier, C. The c-ros
tyrosine kinase receptor controls regionalization and differentiation of epithelial cells in the
epididymis *Genes Dev* **10**, 1184-93 (1996).
- 12. Wellbrock, C. *et al.* Oncogenic BRAF regulates melanoma proliferation through the lineage specific
factor MITF. *PLoS ONE* **3**, e2734 (2008).
- 13. Smith, M.P. *et al.* A PAX3/BRN2 rheostat controls the dynamics of BRAF mediated MITF regulation in
MITF(high) /AXL(low) melanoma. *Pigment Cell Melanoma Res* (2018).
- 14. Berlin, I. *et al.* Phosphorylation of BRN2 modulates its interaction with the Pax3 promoter to control
melanocyte migration and proliferation. *Molecular and cellular biology* **32**, 1237-47 (2012).

REVIEWER COMMENTS

Reviewer #1 (Remarks to the Author):

The authors have made significant changes to their original manuscript in the form of additional clarity of interpretations, transparency of data, and substantial additional experiments and analyses. In doing so, they have substantially strengthened the manuscript and convincingly support each of their main conclusions. I congratulate the authors on the excellent study which I believe both provides both novel and interesting insight to the field as well as a useful new model system and datasets. A few minor edits are requested below.

Minor requests.

1. I still fundamentally disagree with the author's continued choice to omit the gain/amplification category in Fig 1A. It is unethical to selectively omit this category visually from the figure, presumably because it does not cleanly fit with the hypothesis. It is good the authors mention the category in the text, but still, there is no reason to not also add the column visually into the figure for those readers who might internalize the main figures, but miss the sentence in text, and therefore inadvertently misinterpret the data. As the authors allude to in their introduction and discussion, studies of the role of BRN2 in melanoma are wrought with often conflicting observations and confusion, in large part to the practice of "hiding" conflicting data in the supplements, text mention only, or (data not shown). The present study is thorough, excellent and adds substantial clarity to the field due to the careful and nuanced approaches – this reviewer urges the authors to not soil such excellent work through data hiding. Being transparent here will not detract from the overall impact of the study and will, instead, help the field to move past clearly oversimplified binary / linear models and appreciate the nuances and complexities of tumor initiation.

2. Similarly, the analysis presented in Fig S1D does not appear to support the conclusion the authors draw from it as stated in text lines 105-106. The authors present 5 cases where BRN2 was lost in melanoma compared to nevi and 4 cases where it was gained. Thus the statement "It appears that ~30% (5 out of 18) of melanomas presented a mono-allelic loss of BRN2 compared to nevi...we may conclude that this BRN2 mono-allelic loss may occur during the early steps of melanomagenesis" seems misleading when 4 out of those 18 cases had a gain. The analysis is useful and should be kept in the manuscript, but if the 5/18 is highlighted, so too should be the 4/18 for transparency, and the authors should discuss and/or tone down the conclusion.

3. Text line 307 seems to be incomplete: "these results corroborate the immunohistological studies of these tumors. Indeed, (In process)."

Overall, an excellent and important study.

Reviewer #2 (Remarks to the Author):

This is a revised manuscript; the study offers some new insight into the role of Brn2 for melanoma initiation and progression. It would be helpful to reorganize and edited the Manuscript to more accurately describe the data and conclusions.

For example, Fig S1N shows that survival is associated BRN2 loss in the context of monoallelic loss of PTEN without providing a rationale for the reader. Perhaps it would be better to present the data in S1N following the paragraph describing 'Co-occurrence of BRN2 loss with mono-allelic loss of PTEN (line 133)'

Figs. 1C, E: TCGA data retrieved August 2019 should be updated.

It would be interesting to follow-up on several of the gene ontology enrichment process identified by validating them in the mouse models e.g. inflammation, angiogenesis.

The data showing BRN2 is able to directly regulate PTEN transcription are reasonable and novel. The authors also propose that BRN2 is regulating PTEN indirectly via MITF, however they still do not present evidence for BRN2 modulating MITF. Instead, they show that MITF can regulate PTEN (Fig. 6), which is not central to the main conclusions, unless a link between BRN2 and MITF is shown in their models and perform epistatic experiments, for example by evaluating if MITF overexpression could rescue PTEN expression in the context of Brn2 loss.

Other/minor comments

- Clear conclusions summarizing results for each set of experiments will be helpful.
- 1C, E: TCGA data retrieved August 2019 should be updated.
- Analysis of data from precursor nevi/melanoma pairs indicates that mono-allelic loss of BRN2 can occur early in melanomagenesis (5 of 18 samples), as discussed in the results. However, gain of BRN2 occurred in 4 of 18 samples. Please also specify this in the main text.
- If the authors wish to conclude that 'BRN2 promotes the bypass of senescence (by reducing Pten levels)...', this could be experimentally tested by evaluating the % of senescent cells in PTEN+/- + BRN2 WT, BRN2+/, BRN2-/-.
- Several conclusions are stated based on data not shown. If conclusions are important for the overall findings of the study and data is available, it should be included. For example, "The presence or absence of Brn2 did not decrease the ability of these melanoma cell lines to grow in syngeneic mice (data not shown). Based on the data not shown, the authors conclude that "the absence of Brn2 in these melanoma cells does not affect the implantation of the cells on the body wall, the proliferation after their transformation or the induction of angiogenesis in an immunocompetent environment."
- Line 133: first the authors state that: "There was no significant correlation between BRAF or NRAS mutation and BRN2 loss (mono- or bi-allelic), neither in human melanoma samples nor the human cell-line panel (Figure S2A,B)". Then a few lines down (143) they state "found the most frequent genetic constellation that co-occurs with BRN2 loss in melanoma to be BRAFV600X mutation and mono-allelic PTEN loss (Figure S2E)". Please review and edit as appropriate
- Hyperactivation of PI3K/AKT was identified via GSEA; the authors then argue that this was validated based on substantial increased pAKT-S473 and pS6 -S235/236. However, this does not seem to be the case in S4K, M and O where levels of basal pAkt are fairly similar among all genotypes. Please explain.
- Induction of PI3K/AKT should be validated in tumor samples from the different mouse models
- MEK/BRAF inhibitors seem to downregulate BRN2. Brn2 hom loss seems to sensitize cells to PLX treatment, please discuss. Also, It is odd that MAPK + PI3K do not cooperate in the context of Brn2 loss.
- The finding that Brn2 het exhibit Low MITF/High Axl seems to contradict the findings that those cells are more sensitive to MAPKi compared to WT, as the MITFlow/AXI high generally is associated with resistance to BRAF/MEK.
- While the AXL mRNA levels are statistically significantly increased, the magnitude of increase is rather small, except in 501Mel. Additionally, mRNA levels may not necessarily reflect RTK activation. Authors should more accurately state their findings
- Fig. S1J – missing low expression data (red) for CCNC?
- It would be better to use consistent nomenclature for all genes; e.g. BRN2 in Fig 1 vs. POU3F2 in Fig S1
- Line 155: delete inducible, as gives the reader the impression that deletion of Brn2 is inducible.
- Fig. 2, label Pten+/- instead of Pten for clarity as in supplemental figs.
- Replace sacrifice and kill for euthanize

Reviewer #3 (Remarks to the Author):

This reviewer was asked to specifically comment on CUT&RUN part of the manuscript.

The authors will need to address the following comments:

1. There is no CUT&RUN seq description or bioinformatic analysis in the method section, which makes the evaluation impossible.

2. Based on what was presented in Figure 6: In Figure 6C, MITF peaks are even higher in MITF knockout cells compared with WT cells at four distal enhancer peaks. Therefore, the difference of H3K27Ac peaks may be derived from sequencing bias. Input or IgG should be presented to rule out this possibility.

REVIEWER COMMENTS

Reviewer #1 (Remarks to the Author):

The authors have made significant changes to their original manuscript in the form of additional clarity of interpretations, transparency of data, and substantial additional experiments and analyses. In doing so, they have substantially strengthened the manuscript and convincingly support each of their main conclusions. I congratulate the authors on the excellent study which I believe both provides both novel and interesting insight to the field as well as a useful new model system and datasets.

We thank the reviewer for his/her very positive comments and for the constructive criticism that indeed led to a much-improved manuscript.

A few minor edits are requested below.

Minor requests.

1. I still fundamentally disagree with the author's continued choice to omit the gain/amplification category in Fig 1A. It is unethical to selectively omit this category visually from the figure, presumably because it does not cleanly fit with the hypothesis. It is good the authors mention the category in the text, but still, there is no reason to not also add the column visually into the figure for those readers who might internalize the main figures, but miss the sentence in text, and therefore inadvertently misinterpret the data. As the authors allude to in their introduction and discussion, studies of the role of BRN2 in melanoma are wrought with often conflicting observations and confusion, in large part to the practice of "hiding" conflicting data in the supplements, text mention only, or (data not shown). The present study is thorough, excellent and adds substantial clarity to the field due to the careful and nuanced approaches – this reviewer urges the authors to not soil such excellent work through data hiding. Being transparent here will not detract from the overall impact of the study and will, instead, help the field to move past clearly oversimplified binary / linear models and appreciate the nuances and complexities of tumor initiation.

Answer 1. We agree with the Reviewer's suggestion; our intention was not to hide anything, we wanted to simplify the message; however, we recognize the benefits of including all the relevant information. We have added this information in Figure 1A and the corresponding legend. Please see line 600.

2. Similarly, the analysis presented in Fig S1D does not appear to support the conclusion the authors draw from it as stated in text lines 105-106. The authors present 5 cases where BRN2 was lost in melanoma compared to nevi and 4 cases where it was gained. Thus the statement "It appears that ~30% (5 out of 18) of melanomas presented a mono-allelic loss of BRN2 compared to nevi....we may conclude that this BRN2 mono-allelic loss may occur during the early steps of melanomagenesis" seems misleading when 4 out of those 18 cases had a gain. The analysis is useful and should be kept in the manuscript, but if the 5/18 is highlighted, so too should be the 4/18 for transparency, and the authors should discuss and/or tone down the conclusion.

Answer 2. We have modified the sentence to show both sides of the coin, as follows: "It appears that 28% (5 out of 18) or 22% (4 out of 18) of melanomas presented either a mono-allelic loss or a gain of BRN2 respectively compared to nevi (Figure S1D). The situation is clearly complex, but we may conclude that BRN2 mono-allelic loss can occur during the early steps of melanomagenesis." Please see line 105.

3. Text line 307 seems to be incomplete: "these results corroborate the immunohistological studies of these tumors. Indeed, (In process)."

Answer 3. We are sorry for the mistake, it was inserted by error. It has been removed from the manuscript.

Overall, an excellent and important study.

Once again, we would like to thank the reviewer very much for this very positive comment.

Reviewer #2 (Remarks to the Author):

This is a revised manuscript; the study offers some new insight into the role of Brn2 for melanoma initiation and progression.

Answer 1. We thank the Reviewer for recognizing the new insights offered by this manuscript. Indeed, this study reveals the role of BRN2 during melanoma initiation *in vivo*, which had not been demonstrated previously, and it adds some new insight into the role of BRN2 for melanoma progression.

It would be helpful to reorganize and edited the Manuscript to more accurately describe the data and conclusions. For example, Fig S1N shows that survival is associated BRN2 loss in the context of monoallelic loss of PTEN without providing a rationale for the reader. Perhaps it would be better to present the data in S1N following the paragraph describing 'Co-occurrence of BRN2 loss with mono-allelic loss of PTEN (line 133).

Answer 2. This has been modified as suggested by the reviewer. Please see line 142.

Figs. 1C, E: TCGA data retrieved August 2019 should be updated.

Answer 3. In the first submission, we presented the recovered TCGA data in May 2017. In the second submission, we submitted recovered TCGA data in August 2019. Differences between May 2017 and August 2019 were negligible. The differences between August 2019 and February 2021 will undoubtedly be less significant than in the previous update. Indeed, the number of updates performed recently is very small because the vast majority of patients have already died. Therefore, this modification does not seem to be essential, unless the editor wishes us to do so.

It would be interesting to follow-up on several of the gene ontology enrichment process identified by validating them in the mouse models e.g. inflammation, angiogenesis.

Answer 4. This study presented in a supplementary figure (Fig S5) is of course of interest, but this information would not dramatically reinforce the main conclusions of this article, and we feel it would be best left for a follow up manuscript.

The data showing BRN2 is able to directly regulate PTEN transcription are reasonable and novel. The authors also propose that BRN2 is regulating PTEN indirectly via MITF, however they still do not present evidence for BRN2 modulating MITF. Instead, they show that MITF can regulate PTEN (Fig. 6), which is not central to the main conclusions, unless a link between BRN2 and MITF is shown in their models and perform epistatic experiments, for example by evaluating if MITF overexpression could rescue PTEN expression in the context of Brn2 loss.

Answer 5. It is already well established that BRN2 can either positively or negatively regulate transcription of MITF (see for example: ^{1,2}), a key protein of the melanocyte lineage (see introduction). As already mentioned, the role of BRN2 in melanomagenesis has been controversial. Our aim was to show that both BRN2 and MITF can regulate PTEN. Does the activation of PTEN by BRN2 occur at the same time as MITF repressing PTEN? At this point, it is difficult to answer this question, since BRN2 may activate or repress MITF. A better understanding of the regulation of MITF by BRN2 is needed, and it will be necessary to continue to study this specific question, which has been unresolved since 2008, when Brn2 was found to activate or repress the transcription of MITF; likely both scenarios can occur depending on specific conditions or the presence of unidentified co-factors. Here, the indirect regulation of *Pten* by Brn2 via *Mitf* is included as a possibility in the Discussion. To address this question, one option would be to use a genetic approach to delete specifically the binding sites of BRN2 or/and MITF on the *PTEN* locus, or alternatively to modify the activity/amount of BRN2 and/or MITF and evaluate the resulting levels of *PTEN* mRNA. This cannot be included in this article, as these experiments will require the generation of novel mouse models in keeping with the *in vivo* approach used throughout the manuscript. However, we recognize the Reviewer's point and consequently toned down the possible indirect mRNA regulation of *PTEN* by BRN2 via MITF. Please

see line 468.

Other/minor comments

- Clear conclusions summarizing results for each set of experiments will be helpful.

Answer 6. Done, as suggested by the reviewer. Please see lines 105, 127, 146, 368, and 406.

- Analysis of data from precursor nevi/melanoma pairs indicates that mono-allelic loss of BRN2 can occur early in melanomagenesis (5 of 18 samples), as discussed in the results. However, gain of BRN2 occurred in 4 of 18 samples. Please also specify this in the main text.

Answer 7. We now have included a mention of gains of BRN2 in the text, as suggested by the Reviewer (see also reply to Reviewer 1). Please see lines 105.

- If the authors wish to conclude that 'BRN2 promotes the bypass of senescence (by reducing Pten levels)...', this could be experimentally tested by evaluating the % of senescent cells in PTEN+/- + BRN2 WT, BRN2+/, BRN2-/-.

Answer 8. It is known that melanoma initiation is the result of two main cellular processes: initial proliferation and bypass/escape of senescence as mentioned in the introduction. Here, we show that melanoma initiation is induced since the number of individual melanomas is higher when BRN2 is absent (Brn2 F/F) or decreased (Brn2 F/+). In other words, low/null BRN2 induces proliferation, but it is not sufficient to promote melanoma initiation since the production of Braf^{V600E} induces a well-characterized OIS. Moreover, we had mentioned in the text "...BRN2 may act in vivo as an MITF repressor, we have observed that in a non-tumoral context, the specific knock-out of Brn2 in vivo in melanocytes increases the level of Mitf (publication in preparation). In addition, it has been shown that the decrease/loss of PTEN promotes senescence bypass (see for instance Conde-Perez et al., 2015). The lack of BRN2 would contribute to the bypass of senescence by decreasing the levels of PTEN. These physiological and molecular arguments support the hypothesis that the loss/decrease of BRN2 favours the bypass of senescence. We toned down this issue in different sentences. Please see lines 198, 496.

- Several conclusions are stated based on data not shown. If conclusions are important for the overall findings of the study and data is available, it should be included. For example, "The presence or absence of Brn2 did not decrease the ability of these melanoma cell lines to grow in syngeneic mice (data not shown). Based on the data not shown, the authors conclude that "the absence of Brn2 in these melanoma cells does not affect the implantation of the cells on the body wall, the proliferation after their transformation or the induction of angiogenesis in an immunocompetent environment."

Answer 9. Following the Reviewer's recommendation, we now present the results associated with the growth of mouse melanoma cells, mutant or not for Brn2, in syngeneic mice. We included these results as a supplementary table (Table S4) in the text, as follows: "The presence or absence of Brn2 did not decrease the ability of these melanoma cell lines to grow in syngeneic mice (Table S4)." Please see line 235. For this experiment, one hundred thousand cells of each line were subcutaneously injected in C57BL/6 mice. All mice presented tumors from the six tested cell lines (two cell lines per genotype - Brn2 +/+, Brn2 F/+, and Brn2 F/F). This information is included in the Material and methods section of the supplemental information and in the table note.

As to the sentence: "the absence of Brn2 in these melanoma cells does not affect the implantation of the cells on the body wall, the proliferation after their transformation or the induction of angiogenesis in an immunocompetent environment", we included it to simply remind the reader of the significance of cell growth after subcutaneous injection in mice. We modified the text accordingly.

Genotype of the injected cell line	WT	WT	het	het	hom	hom
Name of the cell line	m6	m50	m36	m59	m8	m82
Number of tumor/injected mice	4/4	4/4	4/4	4/4	4/4	4/4

- Line 133: first the authors state that: "There was no significant correlation between BRAF or NRAS mutation and BRN2 loss (mono- or bi-allelic), neither in human melanoma samples nor the human cell-line panel (Figure S2A,B)". Then a few lines down (143) they state "found the most frequent genetic constellation that co-occurs with BRN2 loss in melanoma to be BRAFV600X mutation and mono-allelic PTEN loss (Figure S2E)". Please review and edit as appropriate

Answer 10. The reviewer refers to the supplementary Figure S2. I am not sure I understand the referee's comment. In the first sentence, we do not include the status of PTEN (Figure S2A,B) and in the second sentence we include the status of PTEN (Figure S2E).

More precisely, with the full paragraph in italic "*There was no significant correlation between BRAF or NRAS mutation and BRN2 loss (mono- or bi-allelic), neither in human melanoma samples nor the human cell-line panel (Figure S2A,B).*" This corresponds to the first cited sentence. Then, the reviewer omits to refer to "*We then searched for co-occurring CNAs of other known melanoma-associated genes and found that mono-allelic loss of BRN2 co-occurred with mono-allelic loss of PTEN in approximately 40% of the human melanoma samples in TCGA and in the human cell-line panel (Figure S2C,D). We next evaluated the concomitant BRN2 locus alterations, BRAF/NRAS mutations, and CDKN2A/PTEN alterations and*". then he refers to "*found the most frequent genetic constellation that co-occurs with BRN2 loss in melanoma to be BRAF^{V600X} mutation and mono-allelic PTEN loss (Figure S2E).*"

Taking into account the full context of these sentences, we believe that there is no formal opposition between these statements. To avoid any confusion we modified the text "*BRAF^{V600X} mutation and mono-allelic PTEN loss*" to "*BRAF^{V600X} mutation together with mono-allelic PTEN loss*". Please see line 141.

- Hyperactivation of PI3K/AKT was identified via GSEA; the authors then argue that this was validated based on substantial increased pAKT-S473 and pS6 -S235/236. However, this does not seem to be the case in S4K, M and O where levels of basal pAkt are fairly similar among all genotypes. Please explain.

Answer 11. The referee is right, it is not the case in Figure S4K,M,O: these panels refer to cells grown in 2D-culture in the presence of 10% FCS that are selected for efficient proliferation in vitro, a situation quite different from in vivo settings. We added a comment in the corresponding legend in the supplemental information.

- Induction of PI3K/AKT should be validated in tumor samples from the different mouse models

Answer 12. We already showed that the level of Pten was reduced in BraF-Pten-Brn2-het tumors compared with BraF-Pten-Brn2-WT tumors (Figure S5D). To address the reviewer comments, we presented two sets of data. In Figure S5D, we now present the western blots of WT, het and homozygous tumours for Akt, pAkt, S6, and pS6. Moreover, we evaluated the percentage of cells expressing Pten in BraF-Pten-Brn2-WT compared to BraF-Pten-Brn2-het tumors (Figure 5A). We now included the percentage of cells expressing Pten in BraF-Pten-Brn2-hom tumors. It appears that the percentage of cells producing Pten is higher in BraF-Pten-Brn2-WT melanoma than in BraF-Pten-Brn2-het or hom melanoma. From these results, we may conclude that the number of cells in which the PI3K/AKT signalling pathway is activated is increased in BraF-Pten-Brn2-het or hom melanoma compared to BraF-Pten-Brn2-WT melanoma. We understand that we indirectly answer the reviewer but are convinced that it is sufficient at this point. Please see line 335.

- MEK/BRAF inhibitors seem to downregulate BRN2. Brn2 hom loss seems to sensitize cells to PLX treatment, please discuss. Also, It is odd that MAPK + PI3K do not cooperate in the context of Brn2

loss.

Answer 13. It has been shown that BRAF and PI3K induce BRN2^{3,4}. In consequence, it is expected that the level of BRN2 is decreased in the presence of such inhibitors. In addition, in the absence of BRN2, melanoma cells are indeed more sensitive to PLX treatment (Herbert et al, 2019). In addition, it has been recently reported (Pierce et al., 2020) that human “Melanoma cells expressing higher levels of BRN2 generally showed higher IC50 values compared with those lines with low or no BRN2 expression.” This lower sensitivity with higher BRN2 levels is coherent with the function of BRN2 in DNA repair (Herbert et al., 2019). We added this information in the Discussion. Please see line 563.

We were indeed expecting that MAPK and PI3K would cooperate, but it is not the case in our model system. Deciphering this question is of importance but, not being the main focus of this article, it is best left to a follow-up study.

- The finding that Brn2 het exhibit Low MITF/High Axl seems to contradict the findings that those cells are more sensitive to MAPKi compared to WT, as the MITFlow/AXL high generally is associated with resistance to BRAF/MEK. While the AXL mRNA levels are statistically significantly increased, the magnitude of increase is rather small, except in 501Mel. Additionally, mRNA levels may not necessarily reflect RTK activation. Authors should more accurately state their findings

Answer 14. The reviewer is likely referring to Müller et al (Nature Comm, 2014), Konieczkowski et al (Cancer Discovery, 2014), Tirosh et al (Science, 2016), or Boshuizen et al (Nature Medicine, 2018). For these articles, the experimental settings are very different from ours. They generate *in vivo* or *in cellulo* resistant cells to BRAF/MEK inhibitors and evaluate the level of RNA/proteins including MITF and AXL in sensitive and resistant cells. Indeed, it appears that MITF-low/AXL-high cells are more resistant than normal cells. A large set of gene expression modifications occurred during the selection process, including, e.g., changes in the expression of EGFR or PDGFRb that may affect sensitivity to MAPKi. However, as far as we know, the status of BRN2 was not examined in these studies.

In our case the experimental situation is completely different. Besides the mutations of the Braf and Pten loci, a single mutation was differentially generated at the level of the BRN2 locus, and the cells were not randomly selected for drug resistance, they were naïve. Moreover, the level of Mitf is always low in all Braf-Pten melanoma cells. These key differences make any comparison difficult/impossible.

That said, the referee is right, there is a slight increase of Axl mRNA, but we do not evaluate the activity/amount of Axl protein in these mouse tumors or cell lines since it was not the main focus of this article. We now make the point in the text that the level of AXL is clearly induced in melanoma cell lines in which the level of Axl was originally low (501Mel and SK28). Moreover, we included in the discussion the following sentence “In the future, we will have to evaluate the kinase activity of Axl in this context, and the consequences of its inhibition in cellulo and in vivo to understand the Mitf/Brn2/Axl ménage à trois.” It is now included on line 505.

In general, those are good points, but are not the main focus of the article.

- Fig. S1J – missing low expression data (red) for CCNC?

Answer 15. There were no cases in which CCNC gene-expression was below the 1 TPM threshold we had defined to represent absence of expression. As such this panel is not informative, but we included it for the sake of completion. We made a point of this issue in the figure legend – see supplemental information.

- It would be better to use consistent nomenclature for all genes; e.g. BRN2 in Fig 1 vs. POU3F2 in Fig S1

Answer 16. Done

- Line 155: delete inducible, as gives the reader the impression that deletion of Brn2 is inducible.

Answer 17. The Brn2 deletion is indeed inducible. Tamoxifen has to be added to induce the translocation of CreERt2 to the nucleus to generate the mutation.

- Fig. 2, label Pten+/- instead of Pten for clarity as in supplemental figs.

Answer 18. This has been changed to PtenF/+ (F=flox), as this is the correct genetic notation.

- Replace sacrifice and kill for euthanize.

Answer 19. Modified as requested. It is highlighted in one occasion, see line 209.

- o - 0 - o -

Reviewer #3 (Remarks to the Author):

This reviewer was asked to specifically comment on CUT&RUN part of the manuscript.

The authors will need to address the following comments:

1. There is no CUT&RUN seq description or bioinformatic analysis in the method section, which makes the evaluation impossible.

Answer 1. We thank the reviewer for their comment and have added this information to the supplementary methods.

In brief, all CUT&RUN samples (IgG, anti-H3K27Ac, anti-H3K4Me3 and anti-MITF) were mapped to the human genome (Hg19) using Bowtie2. MACS2 was used to identify significantly enriched peaks (q-value <0.05) using IgG, from the respective cell line (WT or delta-MITF), as background control. Consensus peak files (i.e. peak coordinates) from two independent replicates were generated using the "Bed-intersect" function in Bedtools. Finally, "BAMcoverage" was used to normalize (--RPKM) samples for data visualization in IGV.

2. Based on what was presented in Figure 6: In Figure 6C, MITF peaks are even higher in MITF knockout cells compared with WT cells at four distal enhancer peaks. Therefore, the difference of H3K27Ac peaks may be derived from sequencing bias. Input or IgG should be presented to rule out this possibility.

Answer 2. We again thank the reviewer for their comment and close attention to detail regarding the whole genome CUT&RUN datasets. Indeed, it is possible that sequencing depth discrepancies between samples could result in larger peaks in one sample compared to another, and without a detailed methods section it was impossible for the reviewer to understand how we normalized the data. We apologize for this omission and have amended the text in the supplementary methods. Furthermore, as requested, two tracks representing background IgG control samples from SK28 WT and MITF-knockout cells have been included in Figure 6C.

CUT&RUN experiments performed on SK28 melanoma cells revealed several binding sites for MITF to the PTEN locus using an antibody, from Sigma, directed against MITF. The MITF binding site located at 140kb on the PTEN locus was also revealed by HA antibody ChIP-seq experiments on a second 501mel melanoma cell line stably expressing HA-tagged MITF. Finally, this MITF binding site at the PTEN locus was validated by classical ChIP-qPCR experiments. These three independent experiments and approaches together provide convincing evidence that MITF binds to the PTEN locus."

References

1. Goodall, J. *et al.* Brn-2 represses microphthalmia-associated transcription factor expression and marks a distinct subpopulation of microphthalmia-associated transcription factor-negative melanoma cells. *Cancer Res* 68, 7788-94 (2008).
2. Wellbrock, C. *et al.* Oncogenic BRAF regulates melanoma proliferation through the lineage specific factor MITF. *PLoS ONE* 3, e2734 (2008).
3. Goodall, J. *et al.* The Brn-2 transcription factor links activated BRAF to melanoma proliferation. *Mol Cell Biol* 24, 2923-31 (2004).
4. Bonvin, E., Falletta, P., Shaw, H., Delmas, V. & Goding, C.R. A phosphatidylinositol 3-kinase-Pax3 axis regulates Brn-2 expression in melanoma. *Mol Cell Biol* 32, 4674-83 (2012).

REVIEWERS' COMMENTS

Reviewer #2 (Remarks to the Author):

This authors have addressed the key issues raised by this reviewer. The role of BRN2 in melanomagenesis was evaluated using novel mouse models to establish that BRN2 haplo-insufficiency promotes melanoma initiation and metastasis on a BRAF-mutant/PTEN-heterozygous background. Transcriptional repression of the remaining PTEN allele was elicited by BRN2 knockout/knockdown and shown to be directly regulated by BRN2. The authors have now appropriately stated the conclusions on potential indirect regulation of PTEN by BRN2 via MITF, as regulation of MITF by BRN2 was not experimentally evaluated in these models.

Reviewer #3 (Remarks to the Author):

This reviewer thanks the authors for providing the requested information on Cut & Run and I don't have additional questions.

REVIEWERS' COMMENTS

Reviewer #2 (Remarks to the Author):

This authors have addressed the key issues raised by this reviewer. The role of BRN2 in melanomagenesis was evaluated using novel mouse models to establish that BRN2 haplo-insufficiency promotes melanoma initiation and metastasis on a BRAF-mutant/PTEN-heterozygous background. Transcriptional repression of the remaining PTEN allele was elicited by BRN2 knockout/knockdown and shown to be directly regulated by BRN2. The authors have now appropriately stated the conclusions on potential indirect regulation of PTEN by BRN2 via MITF, as regulation of MITF by BRN2 was not experimentally evaluated in these models.

We thank the reviewer for his/her feedback.

- o - 0 - o -

Reviewer #3 (Remarks to the Author):

This reviewer thanks the authors for providing the requested information on Cut & Run and I don't have additional questions.

We thank the reviewer for his/her feedback.